# Tropospheric and stratospheric wildfire smoke profiling with lidar: Mass, surface area, CCN and INP retrieval

Albert Ansmann[1], Kevin Ohneiser[1], Rodanthi-Elisavet Mamouri[2,3], Daniel A. Knopf[4], Igor Veselovskii[5], Holger Baars[1], Ronny Engelmann[1], Andreas Foth[6], Cristofer Jimenez[1], Patric Seifert[1], and Boris Barja[7]

[1]Leibniz Institute for Tropospheric Research, Leipzig, Germany
[2]Department of Civil Engineering and Geomatics, Cyprus University of Technology, Limassol, Cyprus
[3]ERATOSTHENES Center of Excellence, Limassol, Cyprus
[4]School of Marine and Atmospheric Sciences, Stony Brook University, Stony Brook, NY 11794-5000, USA
[5]Prokhorov General Physics Institute of the Russian Academy of Sciences, Moscow, Russia
[6]Leipzig Institute for Meteorology, University of Leipzig, Leipzig, Germany
[7]Atmospheric Research Laboratory, University of Magallanes, Punta Arenas, Chile

**Correspondence:** A. Ansmann
(albert@tropos.de)

**Abstract.** We present retrievals of tropospheric and stratospheric height profiles of particle mass, volume, surface area, and number concentrations in the case of wildfire smoke layers as well as estimates of smoke-related cloud condensation nucleus (CCN) and ice-nucleating particle (INP) concentrations from backscatter lidar measurements at ground and in space. Conversion factors used to convert the optical measurements into microphysical properties play a central role in the data analysis, besides estimates of the smoke extinction-to-backscatter ratios required to obtain smoke extinction coefficients. The set of needed conversion parameters for wildfire smoke are derived from AERONET observations of major smoke events, e.g., in western Canada in August 2017, California in September 2020, and southeastern Australia in January-February 2020 as well as from AERONET long-term observations of smoke in the Amazon region, southern Africa, and Southeast Asia. The new smoke analysis scheme is applied to CALIPSO observations of tropospheric smoke plumes over the United States in September 2020 and to ground-based lidar observation in Punta Arenas, in southern Chile, in aged Australian smoke layers in the stratosphere in January 2020. These case studies show the potential of spaceborne and ground-based lidars to document large-scale and long-lasting wildfire smoke events in detail and thus to provide valuable information for climate-, cloud-, and air chemistry modeling efforts performed to investigate the role of wildfire smoke in the atmospheric system.

## 1 Introduction

Record-breaking injections of Canadian and Australian wildfire smoke into the upper troposphere and lower stratosphere (UTLS) in 2017 and 2020 caused strong perturbations of stratospheric aerosol conditions in the northern and southern hemisphere. The smoke reached heights up to 23 km (Canadian smoke, 2017) (Hu et al., 2019; Baars et al., 2019; Torres et al., 2020) and more than 30 km (Australian smoke, 2020) (Ohneiser et al., 2020; Kablick et al., 2020; Khaykin et al., 2020), spread over large parts of the stratosphere, and remained detectable for 6-12 months. Smoke particles influence climate conditions

(Ditas et al., 2018; Hirsch and Koren, 2021) by strong absorption of solar radiation and by acting as cloud condensation nuclei (CCN) and ice-nucleating particles (INP) in cloud evolution processes (Engel et al., 2013; Knopf et al., 2018). As discussed by Ohneiser et al. (2021), smoke may have even be involved in the complex processes leading to the record-breaking stratospheric ozone-depletion events in the Arctic and Antarctica in 2020 (CAMS, 2021). Recent studies suggest that such major hemispheric perturbations may become more frequent in future within a changing global climate with more hot and dry weather conditions (Liu et al., 2009, 2014; Kitzberger et al., 2017; Kirchmeier-Young et al., 2019; Dowdy et al., 2019; Jones et al., 2020; Witze, 2020).

Lidars around the world and in space are favorable instruments to monitor and document high altitude aerosol layers in the troposphere and lower stratosphere over long time periods. This was impressively demonstrated after major volcanic eruptions such as the El Chichon and Mt. Pinatubo events (Jäger, 2005; Trickl et al., 2013; Sakai et al., 2016; Zuev et al., 2019). As main aerosol proxies the measured particle backscatter coefficient and the related column-integrated backscatter are used. These optical quantities allow a precise and detailed study of the decay behavior of stratospheric aerosol perturbations. Furthermore, for volcanic aerosol a conversion technique was introduced to derive climate and air-chemistry-relevant parameters such as particle extinction coefficient and related aerosol optical thickness (AOT), mass, and surface area concentration from the backscatter lidar observations (Jäger and Hofmann, 1991; Jäger et al., 1995; Jäger and Deshler, 2002, 2003). Analogously, such a conversion scheme is needed for the analysis of free tropospheric and stratospheric wildfire smoke layers, but is not available yet. The two major stratospheric smoke events in 2017 and 2020 motivated us now to develop a respective smoke-related data analysis concept. The technique covers the retrieval of smoke microphysical properties and the estimation of cloud-relevant aerosol properties such as cloud condensation nucleus (CCN) and ice-nucleating particle (INP) number concentrations. The focus is on backscatter lidar observations at 532 nm, but can easily be extended to 355 and 1064 nm, the other two main laser wavelengths used in atmospheric lidar studies. A preliminary version of the new method was already applied to describe the decay of stratospheric perturbation after the major Canadian smoke injection in the second half year of 2017 (Baars et al., 2019) and in recent studies of stratospheric smoke observed over the North Pole region with ground-based lidar during the winter half year of 2019-2020 (Ohneiser et al., 2021). The retrieval scheme is easy to handle and applicable to lidar observation from ground and in space and thus can also be used to evaluate measurements aquired by the spaceborne CALIPSO (Cloud-Aerosol Lidar and Infrared Pathfinder Satellite Observation) lidar (Winker et al., 2009; Omar et al., 2009; Kar et al., 2019), CATS (Cloud-Aerosol Transport System aboard the International Space Station ISS) (Proestakis et al., 2019), and the Aeolus lidar (Reitebuch, 2012; Reitebuch et al., 2020; Baars et al., 2020; Baars et al., 2021) which continuously monitor the global aerosol distribution.

For completeness, alternative lidar techniques are available to derive microphysical properties of smoke layers from lidar observations (Müller et al., 1999a, 2014; Veselovskii et al., 2002, 2012). These comprehensive inversion methods were successfully applied to wildfire smoke layers in the troposphere (Wandinger et al., 2002; Murayama et al., 2004; Müller et al., 2005; Tesche et al., 2011; Alados-Arboledas et al., 2011; Veselovskii et al., 2015) as well as in the stratosphere (Haarig et al., 2018), and even to a stratospheric volcanic aerosol observation (Mattis et al., 2010). However, this sophisticated approach needs lidar observation at multiple wavelengths of very high quality and is strongly based on directly observed particle extinction coeffi-

cient profiles which are not easy to obtain especially not during the second and final phase of major stratospheric perturbations. The lidar inversion technique can sporadically provide valuable information about the relationship between the optical and microphysical properties of observed aerosol layers and thus can be used to check the reliability of applied AERONET-based conversion factors as shown in Sect. 5.5.

The article is organized as follows. An introduction into the complex chemical, microphysical, morphological, and optical properties of wildfire smoke and the ability of these particles to influence ice formation in clouds is given in Sect. 2. In Sect. 3, we provide an overview of the methodological concept, i.e., the way we derive the microphysical and cloud-relevant smoke properties from height profiles of the particle backscatter coefficient. A central role in the data analysis is played by conversion factors (Mamouri and Ansmann, 2016, 2017). The way we determined the smoke conversion factors from Aerosol Robotic Network (AERONET) (Holben et al., 1998) sunphotometer observations is described in Sect. 4. Section 5 presents the results of the AERONET correlation analysis and the derived set of conversion parameters for fire smoke as obtained from respective observations with AERONET sunphotometers in North America, southern Africa, southern South America, and Antarctica. A summary of the studies and an uncertainty analysis is given in Sect. 6. Case studies of observations of stratospheric Australian smoke with ground-based Raman lidar in Punta Arenas, Chile, in January 2020 and of fresh tropospheric smoke with the CALIPSO lidar over the United States in September 2020 are discussed in Sect. 7. Concluding remarks are given in Sect. 8.

## 2 Wildfire smoke characteristics

The development of a smoke-related conversion method is a difficult task because of the complexity of smoke chemical, microphysical, and morphological properties. To facilitate the discussions in the next sections, a good knowledge of smoke characteristics is necessary and provided in this section. The overview is based on the smoke research and discussions presented by Fiebig et al. (2003); Müller et al. (2005, 2007a); Dahlkötter et al. (2014); China et al. (2015, 2017); Knopf et al. (2018), and Liu and Mishchenko (2018, 2020).

### 2.1 Chemical, physical and morphological properties

First of all, the types of fires, e.g., flaming versus smoldering combustion, the fuel type (burning material) and the combustion efficiency at given environmental and soil moisture conditions determine the initial chemical composition and size distribution of the smoke particles injected into the atmosphere. Burning of biomass at higher temperatures, during flaming fires, generates smaller particles than smoldering fires (Müller et al., 2005). In forest fires, the flaming stage is usually followed by a longer period of smoldering fires.

Smoke particles from forest fires are largely composed of organic material (organic carbon, OC) and, to a minor part, of black carbon (BC). The BC mass fraction is typically <5% (Dahlkötter et al., 2014; Yu et al., 2019), but may reach values of 10-15% in cases of complex mixtures of anthropogenic haze with domestic, forest, and agricultural fire smoke (Wang et al., 2011). Biomass burning aerosol also consists of humic like substances (HULIS) which represent large macromolecules (Mayol-Bracero et al., 2002; Schmidl et al., 2008a, b; Fors et al., 2010; Graber and Rudich, 2006). The particles and released vapors

within biomass burning plumes undergo chemical and physical aging processes during long-range transport. There is strong evidence from lidar observations that smoke particles grow in size during the aging phase (Müller et al., 2007a). Processes that lead to the increase of particle size are hygroscopic growth of the particles, gas-to-particle conversion of inorganic and organic vapors during transport, condensation of large organic molecules from the gas phase in the first few hours of aging, coagulation,

and photochemical and cloud-processing mechanisms. The lidar observations are in agreement with modeling studies of Fiebig et al. (2003) who used the theory of particle aging processes described by Reid and Hobbs (1998). Condensational growth dominates the increase of particle size in the first two days after emission of a plume. Thereafter coagulation in the increasingly diluted plumes becomes the dominating process. A significant shift of the particle size distribution indicated by an increase of the number median radius from about 0.2 $\mu$m shortly after emission to about 0.35 $\mu$m after six days of travel was found

in several cases of Canadian smoke by Müller et al. (2007a). The aging effect has to be considered in the retrieval of smoke conversion factors. We distinguish fresh and aged smoke observations in Sect. 5.

Dahlkötter et al. (2014) analyzed aircraft in situ measurements of a smoke layer advected from North America and observed over Germany at 10-12 km height in September 2011 and found, in agreement with many other airborne in situ observations, an almost monomodal size distribution of smoke particles with a pronounced accumulation mode (particles with diameters

from roughly 200 to about 1400 to 1800 nm). A distinct coarse mode was absent.

The black-carbon-containing smoke particles showed coating thicknesses of roughly 50–220 nm and shell-to-core diameter ratios of typically 2-3. Dahlkötter et al. (2014) assumed a concentric-spheres core-shell morphology for the strongly light-absorbing BC core and further assumed purely light-scattering coating material (i.e., no absorption by the shell) in their analysis of the airborne in situ observations. The authors emphasized that their core-shell model is an idealized scenario because the

BC cores of combustion particles are fractal-like or compact aggregates and BC can be mixed with light-scattering material in different ways, including, e.g., surface contact of BC with the light-scattering components, full immersion of BC in the light-scattering component or immersion of the light-scattering components in the BC aggregate. A process that can produce near-surface BC morphology is coagulation of almost bare BC aggregates with BC-free particles. Condensation of secondary organic or inorganic aerosol components on BC particles can either result in particles with core-shell morphology (concentric

or eccentric) or with near-surface BC morphology. All these possible morphology features must be considered in the discussion and estimation of the smoke optical properties and of the potential of smoke particles to serve as INP (Sect. 2.2 and 3.1).

Changes in the morphology (size, shape, and internal structure) of smoke particles and their internal mixing state (e.g., soot particle coating) are ongoing during long-range transport. As China et al. (2015) pointed out, freshly emitted soot particles, i.e., BC particles, are typically hydrophobic, lacy fractal-like aggregates of carbonaceous monomers and become hydrophilic

as a result of coating and other aging processes. Lace soot undergoes compaction upon humidification. All these effects lead to an increased ability of smoke particles to serve as CCN with increasing long-range travel time.

Soot compaction (and collapse of the core structures) changes also the scattering and absorption cross sections depending on the refractive index, the monomer diameter, and the structural details. Many publications dealing with the optical properties became available in recent years (China et al., 2015; Liu and Mishchenko, 2018, 2020; Kahnert, 2017; Yu et al., 2019; Gialitaki

et al., 2020). Liu and Mishchenko (2018) mentioned that their model considers eleven different model morphologies ranging

from bare soot to completely embedded soot–sulfate and soot–brown carbon mixtures. In agreement with earlier studies, they found that for the same amount of absorbing material, the absorption cross section of internally mixed soot can be more than twice that of bare soot. Thus absorption increases as soot accumulates more coating material during long-range transport. As a general finding of the modeling studies, the absorption enhancement is a complex function of many factors such as the size
and shape of the soot aerosols, the mixing state, the location of soot within the host, and the amount and composition of the coating material. All these facts make it necessary to distinguish between fresh smoke (<2.5 days after injection) and aged wildfire smoke (>2.5 days of long-range transport) in our attempt to determine smoke conversion parameters.

## 2.2   Cloud-relevant properties

As already mentioned, smoke particles after long-range transport seem to be favorable CCN because they become increasingly
hydrophilic during aging. In contrast to the impact of smoke on cloud droplet formation, the characterization of their influence on ice nucleation is rather difficult. The link between ice nucleation efficiency and particle chemical and morphological properties and the ongoing modifications of the properties during long-range transport is largely unresolved (China et al., 2017). However, it is widely assumed that the ability of smoke particles to serve as INP mainly depends on the organic material (OM) in the shell of the coated smoke particles (Knopf et al., 2018). BC is not considered to be an important contributor to immersion
freezing (Möhler et al., 2005; Ullrich et al., 2017; Schill et al., 2020; Kanji et al., 2020) which is assumed to be the preferred heterogeneous ice nucleation mode.

Knopf et al. (2018) present a review on the role of organic aerosol (OA) and OM in atmospheric ice nucleation. A unique feature of OA particles is that they can be amorphous and can exist in different phases, including liquid, semisolid, and solid (or glassy) states in response to changes in temperature (T) and relative humidity (RH) (Koop et al., 2011; Zobrist et al., 2008;
Knopf et al., 2018). At low temperatures, e.g., in the UTLS region, where the atmospheric temperature can be as low as 180 K, it is conceivable to assume that the particles are in a glassy state. Most of the secondary organic aerosol particles are solid above 500 hPa (about 5 km) according to modeling studies and for temperatures <240 K (Shiraiwa et al., 2017).

It has been shown that humic and fulvic matter can act as deposition nucleation and immersion freezing INPs (Wang and Knopf, 2011; Rigg et al., 2013; Knopf and Alpert, 2013; Knopf et al., 2018). Furthermore, these macromolecules can undergo
amorphous phase transition under typical tropospheric conditions (Wang et al., 2012; Slade et al., 2017) similar to the processes we assume the organic coating of the smoke particles experience.

Aerosol particles serving as INPs usually provide an insoluble, solid surface that can facilitate the freezing of water (Knopf et al., 2018). Deposition ice nucleation is defined as ice formation occurring on the INP surface by water vapor deposition from the supersaturated gas phase. Though, recent studies suggest that deposition ice nucleation can be the result of pore
condensation freezing, where homogeneous ice nucleation occurs at lower supersaturation in nanometer-sized pores (David et al., 2019; Marcolli, 2014). When the supercooled smoke particle takes up water or its shell deliquesces, immersion freezing can proceed, where the INP immersed in an aqueous solution can initiate freezing (Knopf et al., 2018; Berkemeier et al., 2014). Finally, if the smoke particle becomes completely liquid (and no insoluble part within the particle is left), homogeneous freezing will take place at temperatures below  235 K (Koop et al., 2000).

However, in reality, at given air mass lifting conditions, the ice nucleation process can be very complex. The time that solid OM needs for transition to a more liquid state, termed as humidity-induced amorphous deliquescence, can range from several minutes to days at temperatures low enough for ice formation (Mikhailov et al., 2009; Berkemeier et al., 2014; Knopf et al., 2018). Thus the phase change (as function of T and RH) can be longer than typical cloud activation time periods (governed

by the updraft velocity), potentially inhibiting full deliquescence and allowing the OA or the organic coating to serve as INP. When amorphous OA or OM are involved in ice nucleation, the condensed-phase diffusion processes within OA particles will most probably govern the ice nucleation pathway (Wang et al., 2012).

The following potential scenarios of atmospheric ice nucleation are uniquely attributable to the presence of amorphous OM: (1) Ice formation in the glassy region may be due to ice nucleation on the solid organic particle, i.e., deposition ice nucleation.

(2) During partial deliquescence, a residual solid core is coated by an aqueous shell, and immersion freezing may proceed. (3) At full deliquescense RH, the particles are completely liquid (and contain no solid soot fragments), homogeneous freezing will occur at temperatures below about 238 K. (4) The presence of a glassy phase in disequilibrium with surrounding water vapor (e.g., cloud activation at fast updrafts as discussed below) may suppress or initiate ice nucleation beyond the homogeneous ice nucleation limit (Berkemeier et al., 2014; Knopf et al., 2018). A slower updraft velocity allows for more time for deliquescence

to proceed, potentially resulting in full deliquescence of the OA particle at warmer and drier conditions compared to when a faster updraft is active. Therefore, the same OM can be present in different phase states under the same atmospheric thermodynamic conditions (i.e., $T$ and relative humidity over ice $RH_i$), resulting in different ice nucleation pathways and corresponding ice nucleation rates. OA particle size or coating thickness can also impact the rate and atmospheric altitude of the organic phase change, as larger particles or thicker coatings require more time to reach full deliquescence (Charnawskas et al., 2017) . There

are many more peculiarities of amorphous OM that makes INP parameterization and prediction efforts very complicated as discussed in detail by Knopf et al. (2018).

Since amorphous smoke OA may take up water and partially deliquesce resulting in an aqueous solution at possibly subsaturated conditions, we apply the water-activity based immersion freezing (ABIFM) parameterization (Knopf and Alpert, 2013; Alpert and Knopf, 2016) and homogeneous ice nucleation parameterization by Koop et al. (2000). ABIFM derives the number

of INPs per volume of air for a given time period, when $T$, $RH$, and particle surface area $s$ are known (see Sect. 3.1.1). A deposition ice nucleation scheme based on classical nucleation theory is outlined in addition (Sect. 3.1.3) to cover the potential pathway of glassy smoke particles to serve as INPs. Again, $T$, $RH$, and $s$ are input in the INP estimation.

To demonstrate the prediction or retrieval of smoke INP profiles from lidar observations in Sect. 7, we apply exemplary two OA model systems serving as surrogates of amorphous organic smoke particles. One is based on a macromolecular humic

or fulvic acid that undergoes amorphous phase transitions in response to changes in $RH$ and $T$ (Wang et al., 2012) and free troposphere long-range transported particles that possess an organic coating acting as INPs (China et al., 2017).

## 3 Methodological background: Microphysical properties from backscatter coefficients

The goal of the study is to provide a set of conversion parameters that permits the estimation of smoke microphysical properties from particle backscatter coefficients measured at 532 nm. A smoke observation with ground-based lidar at Punta Arenas, in southern Chile, is shown in Figure 1 (Ohneiser et al., 2020). We will use this measurement as case study in Sect. 7.1 and will apply all conversion procedures to this observation.

The methodological background of the conversion of optical into microphysical particle properties is given by Mamouri and Ansmann (2016, 2017). It is out of the scope of this article to present a detailed approach how an aerosol layer can be unambiguously identified and classified as a smoke layer. In case of single-wavelength backscatter lidars, backward trajectory analysis is the main tool to identify smoke layers and link them to the most probable fire source region. In case of modern aerosol lidars, equipped with polarization-sensitive channels and aerosol and molecular backscatter channels at several wavelengths, favorable conditions are given to identify smoke layers based on the complex set of available information on particle backscatter and extincton coefficients, depolarization ratio, and lidar ratio (Wandinger et al., 2002; Müller et al., 2005; Tesche et al., 2011; Burton et al., 2012, 2015; Giannakaki et al., 2015; Giannakaki et al., 2016; Prata et al., 2017; Haarig et al., 2018; Hu et al., 2019; Adam et al., 2020; Ohneiser et al., 2020; Ohneiser et al., 2021). However, an unambiguous and accurate quantification of the smoke fraction or contribution to the measured optical backscatter and extinction properties and the separation of smoke and soil dust fractions, remains difficult. Soil dust may have been injected together with the smoke by the hot fires.

Regarding the separation of smoke and dust fractions by means of the polarization lidar technique (Tesche et al., 2009, 2011; Nisantzi et al., 2014), we have to distinguish two branches. As long as the smoke-containing layers occur at low altitudes (in the lower and middle troposphere up to 5-7 km height), we can apply the traditional approach to determine the smoke fraction in dust-smoke mixtures by assuming a low smoke depolarization ratio of <0.05 and a high mineral dust depolarization ratio of 0.31. In the lower and middle troposphere, aging of the smoke particles is usually fast including the development of a spherical shape of the aged smoke particles. Furthermore, most of the smoke particles are liquid (at least the shell) at comparably high temperatures and moisture levels. All this leads to a low smoke depolarization ratio at all laser wavelengths from 355 to 1064 nm (Haarig et al., 2018).

However, if the smoke is lifted directly into the upper troposphere and lower stratosphere (UTLS), the smoke properties and aging features may be significantly different. With increasing height, and thus decreasing temperature, water vapor content, and amount of condensable gases, the aging process is slow and the smoke particle become partly glassy. These effects seem to prohibit the development of a perfect spherical shape of the shells. As a consequence, the depolarization ratio can be as high as 0.15-0.2 at 532 nm at greater heights (Burton et al., 2015; Haarig et al., 2018; Hu et al., 2019; Ohneiser et al., 2020). However, we also observed low smoke depolarization ratios in the UTLS region (Ohneiser et al., 2021). Thus, in the case of UTLS smoke observations, the dust-smoke separation technique cannot be used. We have to assume that smoke layers are dominated by smoke (smoke fraction >0.9) in the UTLS regime, and the soil dust fraction can be neglected at these heights.

To obtain height profiles of smoke in terms of volume concentration $v(z)$, surface area concentration $s(z)$, particle number concentrations $n_{50}(z)$, considering all particles with radius > 50 nm, and the large-particle number concentration $n_{250}(z)$,

considering particles with particle radius >250 nm, we have the following four basic relationships:

$$v(z) = c_{\mathrm{v}} L\beta(z),\qquad(1)$$

$$s(z) = c_{\mathrm{s}} L\beta(z),\qquad(2)$$

$$n_{250}(z) = c_{250} L\beta(z),\qquad(3)$$

$$n_{50}(z) = c_{50}\left[L\beta(z)\right]^{x}\qquad(4)$$

with the particle backscatter coefficient $\beta(z)$ at height $z$ and the extinction-to-backscatter or lidar ratio $L$. The needed conversion factors $c_{\mathrm{v}}$, $c_{\mathrm{s}}$, $c_{250}$, and $c_{50}$, and the extinction exponent $x$ for 532 nm are obtained from the analysis of AERONET observations during situations dominated by wildfire smoke. The results of our smoke-related AERONET data analysis is presented in Sect. 5.

An important input parameter is the smoke lidar ratio $L$, required to obtain the smoke extinction coefficient $\sigma = L\beta$ in the first step of the conversion procedure. As discussed in the review of Adam et al. (2020), the smoke lidar ratio can vary from 25 to 150 sr at 532 nm. However, most studies show that the 532 nm lidar ratio is typically in the range of 70 sr$\pm$25 sr. For 355 nm, lidar ratios were mostly found around 75$\pm$25 sr for fresh smoke and 55$\pm$20 sr for aged smoke. Table 1 provides an overview of the large range of smoke lidar ratios. Aged smoke shows a characteristic $L$ ratio of $L_{355\ \mathrm{nm}}/L_{532\ \mathrm{nm}} < 1$. This feature allows

a clear unambiguous identification of smoke layers after long-range transport (Müller et al., 2005; Noh et al., 2009; Nicolae et al., 2013; Ohneiser et al., 2020). The reason for the large spectrum of lidar ratios are the complex smoke properties (size, shape, composition) as discussed in Sect. 2. Extended discussions on smoke lidar ratios can be found in Nicolae et al. (2013), Haarig et al. (2018), and Adam et al. (2020).

    We recommend to use a lidar ratio of 55 sr for 355 nm and 70 sr for 532 nm for aged smoke if there is no possibility to obtain

actual lidar ratio information from Raman lidar (Wandinger et al., 2002; Veselovskii et al., 2015; Haarig et al., 2018; Ohneiser et al., 2020; Ohneiser et al., 2021) or High Spectral Resolution Lidar (HSRL) observations (Wandinger et al., 2002; Burton et al., 2015), or in the way Prata et al. (2017) proposed in the case of the CALIPSO lidar to estimate the lidar ratio of smoke layers embedded in clear air. For fresh smoke, an appropriate value for the lidar ratio seems to be 70-80 sr at both wavelengths.

    From the obtained values of $v$, $s$, and $n_{50}$ further relevant parameters can be calculated. The smoke mass concentration $m$ is

given by

$$m(z) = \rho v(z),\qquad(5)$$

with $\rho$ the density of the smoke particles. Li et al. (2016) investigated different smoke aerosols in the laboratory by burning of different straw types and found densities of 1.1 to 1.4 g cm$^{-3}$ for the produced smoke particles. For organic particles $\rho_{\mathrm{OM}}$ was about 1.05$\pm$0.15 g cm$^{-3}$ and for $\rho_{\mathrm{EC}}$ (elemental carbon) they yielded 1.8 g cm$^{-3}$. Chen et al. (2017) reviewed the smoke

research in China and concluded that the smoke particle density is 1.0-1.9 g cm$^{-3}$. Thus in cases with 2-10% of BC the overall smoke particle density should be in the range of 1.0–1.3 g cm$^{-3}$.

    The particle concentration $n_{50}$ is a good aerosol proxy for aerosol particles serving as cloud condensation nuclei (CCN),

$$n_{\mathrm{CCN},S_{\mathrm{w}}=0.2\%}(z) = n_{50}(z).\qquad(6)$$

The CCN concentration is a strong function of updraft speed and thus water supersaturation $S_w$. The number concentration $n_{50}$ roughly indicates the CCN concentration for weak updrafts and frequently observed low water supersaturations of $S_w = 0.2\%$. Water supersaturation values may be in the range of 0.4–0.7% in strong updrafts. Then the CCN concentrations is a factor of about 2 higher than $n_{50}$.

In the case of free-tropospheric and stratospheric smoke, we assume that the relative humidity in the smoke plumes is typically <60% so that the derived $n_{50}$ values represent the number concentrations for dry aerosol particles, required in the CCN estimation. The estimation of CCN concentration in cases with high relative humidity and corresponding aerosol water-uptake effects is described in Mamouri and Ansmann (2016).

The particle concentration $n_{250}$ indicates the reservoir of favorable INPs and is even used as input in dust-INP parame-terisiations (DeMott et al., 2015). However, in the case of smoke the input parameter in the INP retrieval is the surface area concentration $s$,

$$n_{\mathrm{INP}}(z) = f(s(z), S_i(z), T(z)). \tag{7}$$

The INP concentration is a function of $s$, the ice supersaturation $S_i$ (which occurs during lifting processes), and temperature $T$. Details of the complex INP parameterization are given in Sect. 3.1.

Finally, information on smoke particle number concentrations ($n_{50}$, $n_{250}$) and surface area concentration $s$ at stratospheric heights is of interest in studies of heterogeneous formation of polar stratospheric clouds (PSCs). A significant increase in smoke aerosol particle concentration may have a sensitive impact on the evolution of PSCs and their microphysical properties (Voigt et al., 2005; Hoyle et al., 2013; Engel et al., 2013; Zhu et al., 2015).

In order to use the developed smoke retrieval formalism presented here in the case of backscatter lidars operated at single wavelengths of $\lambda$=355 nm or 1064 nm backscatter lidars, we need to estimate the respective backscatter coefficient at 532 nm in the first step. The 532 nm backscatter profiles within smoke layers may be estimated by using typical smoke color ratios $\beta(532\,\mathrm{nm})/\beta(\lambda)$. This aspect is further discussed in Sect. 6.

## 3.1 INP parameterization

As discussed in Sect. 2.2, the estimation of INP concentrations is challenging due to the chemical complexity of the smoke aerosol. The parameterizations introduced in this section cover the OM-related ice nucleation for the temperature range in the upper troposphere ($< -40°$C). Only for these low temperatures, organic smoke particles may be able to influence ice nucleation in the atmosphere. In the following, we present procedures to compute INP concentrations for immersion freezing, deposition ice nucleation, and homogeneous freezing.

### 3.1.1 Immersion freezing

Organic smoke particles that have undergone long-range transport are chemically complex and INP paramterizations that cap-ture the ice formation rate at upper tropospheric and lower stratospheric conditions (i.e., including subsaturated conditions) are scarce (Knopf et al., 2018). Knopf and Alpert (2013) introduced the water-activity-based immersion freezing model ABIFM,

drawn from the water-activity-based homogeneous ice nucleation theory (Koop et al., 2000). Knopf and Alpert (2013) present an ABIFM parameterization for two types of humic compounds based also on experimental data by Rigg et al. (2013) that is valid for saturated and subsaturated atmospheric conditions. For demonstration of our method, we chose to apply the ABIFM for Leonardite (a standard humic acid surrogate material) to represent the amorphous organic coating of smoke particles. The

ABIFM allows prediction of the ice particle production rate $J_{\mathrm{het,I}}$ as a function of ambient air temperature $T$ (freezing temperature), ice supersaturation $S_{\mathrm{i}}$, particle surface area $s$, and time period $\Delta t$ for which a certain level of ice supersaturation $S_{\mathrm{i}}$ is given. For demonstration purposes, we simply assume a constant supersaturation period $\Delta t$ of 10 minutes (600 s). Such supersaturation conditions may occur during the upwind phase of a gravity wave.

According to Eqs. (6)-(8) in Alpert and Knopf (2016), we calulate the so-called water activity criterion (Koop et al., 2000)
in the first step:

$$\Delta a_{\mathrm{w}} = a_{\mathrm{w}} - a_{\mathrm{w,i}}(T)\,. \tag{8}$$

The term $a_{\mathrm{w,i}}$ in Eq. (8),

$$a_{\mathrm{w,i}} = P_{\mathrm{i}}(T)/P_{\mathrm{w}}(T) \tag{9}$$

is the ratio of ice saturation pressure $P_{\mathrm{i}}$ to water saturation pressure $P_{\mathrm{w}}$ as function of temperature $T$ and can be accurately
determined by using Eq. (7) in Koop and Zobrist (2009). When condensed-phase and vapor phase are in equilibrium, the water activity $a_{\mathrm{w}}$ is equal to $RH_{\mathrm{w}}$ (written as 0.75 if $RH_{\mathrm{w}}$=75%) in the air parcel in which ice nucleation takes place (e.g., in a cirrus layer at height $z$ at temperature $T$). Relative humidity and temperature values may be available from radiosonde ascents or taken from data bases with re-analyzed global atmospheric data. However, the actual $RH_{\mathrm{w}}$ and $T$ values during the lifting process (associated with cooling and increase in $RH_{\mathrm{w}}$ and decrease in $T$ in the air parcel) remain always unknown and need
to be estimated in the studies of a potential smoke impact on cirrus formation. The organic aerosol type Leonardite needs a relative humidity over ice $RH_{\mathrm{i}}$ of about 130% or $\Delta a_{\mathrm{w}} = 0.2$ at $-50°C$ to become efficiently activated as INP.

In the next step, the ice crystal nucleation rate coefficient $J_{\mathrm{het,I}}$ (in $\mathrm{cm}^{-2}\,\mathrm{s}^{-1}$) is calculated:

$$\log_{10}(J_{\mathrm{het,I}}) = b + k\Delta a_{\mathrm{w}}\,. \tag{10}$$

The particle parameters $b$ and $k$ are determined from laboratory studies for different organic aerosol material. Table 2 contains
the parameters for two different natural organic substances (Pahokee Peat and Leonardite) (Knopf and Alpert, 2013) which serve as surrogates of the organic coating of the atmospheric smoke particles. Leonardite, an oxidation product of lignite, is a humic-acid-containing soft waxy particle (mineraloid), black or brown in color, and soluble in alkaline solutions. Both substances served as surrogates for humic-like substances (HULIS, Sect. 2.1) in extended immersion freezing laboratory studies (Knopf and Alpert, 2013; Rigg et al., 2013). Organic aerosols containing HULIS are ubiquitous in the atmosphere. We also
applied the ABIFM parameterization to aerosol samples representing free tropospheric aerosol (FTA, China et al., 2017) collected on substrates on the Azores for off-line micro-spectroscopic single-particle analysis and ice nucleation experiments. According to backward trajectories, the air masses arriving at the Azores crossed western parts of North America during the

main fire season (August-September). FTA showed clear smoke signatures. Note that Eq. (10) delivers strongly fluctuating solutions of $J_{\mathrm{het,I}}$ when $\Delta a_{\mathrm{w}}$ is small, and robust, less fluctuating $J_{\mathrm{het,I}}$ values for $\Delta a_{\mathrm{w}} > 0.1$.

In the final step, we obtain the number concentration of smoke INP for the immersion freezing mode,

$$n_{\mathrm{INP,I}} = s J_{\mathrm{het,I}} \Delta t \qquad (11)$$

with the surface area concentration $s$ of the smoke particles in $\mathrm{cm^2\ m^{-3}}$ and the time period $\Delta t$ (in seconds) for which constant or almost constant ice supersaturation conditions are given. This can be the time period of a short updraft event (of a few minutes, 120-300 s) or of the lifting period of a gravity wave ($>600$ s). Long lasting lifting phases of gravity waves can be up to 20 minutes (1200 seconds) as our Doppler lidar and radar observations conducted in several field campaigns during the last 10 years indicate.

### 3.1.2 Homogeneous freezing

Alternatively to smoke particles acting as heterogeneous INPs, we need to consider full deliquescence of smoke particles so that homogeneous freezing comes into play. Following Koop et al. (2000), the ice nucleation rate coefficient for homogeneous freezing is obtained from

$$\log_{10}(J_{\mathrm{hom}}) = -906.7 + 8502\Delta a_{\mathrm{w}} - 26924(\Delta a_{\mathrm{w}})^2 + 29180(\Delta a_{\mathrm{w}})^3 \qquad (12)$$

for $0.26 < \Delta a_{\mathrm{w}} < 0.34$. The INP concentration is then obtained from

$$n_{\mathrm{INP,hom}} = v J_{\mathrm{hom}} \Delta t \qquad (13)$$

with the particle volume concentration $v$ in $\mathrm{cm^3\ m^{-3}}$. Homogeneous freezing proceeds at $RH_{\mathrm{i}} \approx 150\%$ at $-50°\mathrm{C}$ (i.e., $\Delta a_{\mathrm{w}} \approx 0.31$), whereas 130% ($\Delta a_{\mathrm{w}} = 0.2$) are required at $-50°\mathrm{C}$ to activate Leonardite-containing particles. Thus at slow ascend conditions heterogeneous ice nucleation on smoke particles may dominate ice formation in cirrus layers.

### 3.1.3 Deposition nucleation

Wang and Knopf (2011) provide a simplified parameterization of deposition ice nucleation (DIN) based on classical nucleation theory that describes the DIN efficiency of humic and fulvic acid compounds as a function of ambient temperature $T$ and the humidity parameters $RH_{\mathrm{i}}$ and $S_{\mathrm{i}}$. An alternative DIN parameterization is provided by, e.g., Hoose et al. (2010). A detailed description of the approach presented here is given in Sect. 3.6 in Wang and Knopf (2011) and thus only a brief introduction is given in the following.

The INP efficiencies are expressed as a function of the contact angle $\Theta$ which describes the relationship of surface free energies among the three involved interfaces including water vapor, ice embryo, and INP. $\Theta$ is parameterized as a function of $RH_{\mathrm{i}}$ (Eq. (8) in Wang and Knopf (2011)).

The compatibility parameter $m_{\Theta} = \cos(\Theta)$ (expressing the match between ice embryo and INP) is then used to determine the so-called geometric factor $f_{\mathrm{g}}(m_{\Theta})$ (Eq. (7) in Wang and Knopf (2011)), the free energy of ice embryo formation

$\Delta F_{\text{g,het}}(f_{\text{g}}, T, S_{\text{i}})$ (Eq. (6) in Wang and Knopf (2011)), and finally the ice crystal nucleation rate $J_{\text{het,D}}$ (Eq. (5) in Wang and Knopf (2011)) in $\text{cm}^{-2}\,\text{s}^{-1}$,

$$J_{\text{het,D}} = 10^{25} \exp\left(\frac{-\Delta F_{\text{g,het}}}{k_{\text{B}} T}\right) \tag{14}$$

with the Boltzmann constant $k_{\text{B}}$. The final step is then:

$$n_{\text{INP,D}} = s J_{\text{het,D}} \Delta t. \tag{15}$$

In terms of the contact-angle-based approach, $\Theta = 180°$ represents the case of homogeneous ice nucleation. The smaller $\Theta$, the greater the propensity of the INP to act as deposition nucleation INP.

At the end of this section it remains to be emphasize that we put together several INP parameterizations in Sect. 3.1 for demonstration purposes. The research on the smoke impact on atmospheric ice formation is ongoing (Knopf et al., 2018). Presently, uncertainties in the prediction of $J_{\text{het,I}}$ and $J_{\text{het,D}}$ for organic aerosols are very high (Wang and Knopf, 2011; China et al., 2017). However, the procedures introduced above allow us to estimate INP concentration profiles for organic aerosols and to study the potential impact of wildfire smoke on ice formation in tropospheric mixed-phase and ice clouds. In the upcoming years, strong field activities are required including comparisons of airborne in situ with lidar observations of smoke INP concentrations as successfully performed in the case of Saharan dust (Schrod et al., 2017; Marinou et al., 2019) and so-called cirrus closure experiments as realized in the case of cirrus formation in pronounced Saharan dust layers (Ansmann et al., 2019b) in order to check the applicability of developed smoke INP parameterizations and to quantify the uncertainties in the INP estimates under real-world meteorological, cloud, and aerosol conditions. A first closure study with respect to smoke-cirrus interaction was recently presented by Engelmann et al. (2020).

## 4 AERONET sites and data analysis

The AERONET data base (AERONET, 2021) contains unique multiyear climatological data sets of spectrally resolved aerosol optical properties and related underlying microphysical properties of aerosol particles (e.g., size distribution, volume and surface area concentration). These AERONET products are available in the data base for purely marine, dust, biomass-burning smoke, and anthropogenic haze conditions as well as for complex mixtures of these basic aerosol types. We used the advantage of the AERONET data base already to derive the conversion parameters for marine and Saharan dust conditions (Mamouri and Ansmann, 2016, 2017) and extended the dust-related study later on to many desert dust regions around the world (Ansmann et al., 2019a). Now, we apply the methodology to the wildfire aerosol type.

### 4.1 AERONET sites

The smoke conversion parameters $c_{\text{v}}$, $c_{\text{s}}$, $c_{50}$, $c_{250}$, and $x$, required to solve Eqs. (1) - (4), were determined from sunphotometer observations at nine AERONET stations, distributed over several continents. Figure 2 shows the considered AERONET stations. The observations at these sites cover the full range of smoke scenarios, from fresh to aged plumes, for different fire types and burning material, and smoke occurrence in the troposphere and stratosphere.

Yellowknife (AERONET site: Yellowknife-Aurora) and Churchill in Canada were selected because these AERONET sites were located in the outflow region of major smoke plumes which originated from the record-breaking wildfires in British Columbia (Hu et al., 2019; Baars et al., 2019; Torres et al., 2020), Canada, in August 2017. Strong pyrocumulonimbus (pyroCb) towers (Fromm et al., 2010) developed and lifted enormous amounts of wildfire smoke into the upper troposphere and lower stratosphere (UTLS) from 21:00 UTC on 12 August to 00:30 UTC on 13 August 2017 (Peterson et al., 2018). The smoke observation at Yellowknife and Churchill could be thus well assigned to the time after injection, and allowed us to study the change of the smoke conversion parameters as a function of time from 12-18 hours to about 5 days after injection.

The AERONET stations at Rio Gallegos (CEILAP-RG), Argentina, Punta Arenas (Punta-Arenas-UMAG), Chile, at the southernmost tip of South America, and Marambio in Antarctica were selected because well aged smoke layers crossed these stations in January and February 2020 (Ohneiser et al., 2020). The smoke originated from strong fires in southeastern Australia and travelled the 10000 km distance within 8-12 days. Strong pyroCb activity lifted the smoke layers up to UTLS heights and self-lifting processes (Boers et al., 2010) caused further ascent to heights 10-20 km above the tropopause (Ohneiser et al., 2020; Kablick et al., 2020; Khaykin et al., 2020). The background AOT levels are clearly below 0.05 at 532 nm at these high northern and southern mid-latitudinal stations, far away from industrialized centers, so that the smoke layers could be clearly identified and dominated the sunphotometer observations over many days (Yellowknife, Churchill) and weeks (Punta Arenas, Rio Gallegos, Marambio).

In order to consider several centers of biomass burning of global importance we selected six further AERONET stations. Smoke from exceptionally strong forest fires in the western United States and western Canada was observed over Reno (Univ-of-Nevada-Reno), Nevada, and Table Mountain (Table-Mountain-CA), California, from end of August to mid October 2020 (in close distance to the fire sources) and allowed the determination of conversion parameters for very fresh and mixtures of fresh and aged North American tropospheric smoke layers.

We downloaded long-term observations performed at the AERONET stations Alta Floresta, Brazil (Amazonian forest fires), Mongu, Zambia in southern Africa, Mukdahan, Thailand, and Singapore in Southeast Asia to consider observations in key fire areas of global importance. The Mongu data sets consists of sunphotometer observations at the Mongu site from 1997-2009 and at the Mongu-Inn site from 2013-2019. Fairly constant burning conditions are given at Mongu from July to November of each year. The long-term observations in the Amazon region, southern Africa, and Southeast Asia cover smoldering and flaming fires, fresh and aged smoke layers, as well as agricultural, grassland, savannah, peat, forest, and bush fires. The selection of these AERONET stations in key burning areas was guided by the smoke study of Sayer et al. (2014).

The AERONET smoke studies are supplemented by multiwavelength lidar observations of smoke conversion parameters. These vertically resolved observations were performed at Punta Arenas (Ohneiser et al., 2020), Manaus, Brazil (Baars et al., 2012), near Washington, DC (Veselovskii et al., 2015), at Cabo Verde, in the outflow regime of central western African smoke (Tesche et al., 2011), at Leipzig and Lindenberg, Germany (Wandinger et al., 2002; Haarig et al., 2018), and on the German icebreaker Polarstern drifting through the high Arctic close to the North Pole during the winterhalf year of 2019-2020 (Engelmann et al., 2020; Ohneiser et al., 2021). The lidar results are shown in Sect. 5.5. The retrieval of the microphyscial properties was based on backscatter coefficients measured at 355, 532, and 1064 nm and extinction values at 355 and 532 nm (Müller

et al., 1999a, b; Veselovskii et al., 2002), except for the smoke observations over Lindenberg in the summer of 1998. Here, particle backscatter coefficients at six wavelength (355, 400, 532, 710, 800, 1064 nm) and extinction coefficients at 355 and 532 nm were available (Wandinger et al., 2002).

## 4.2 AERONET data analysis

We used the version-3 level-2.0 inversion AERONET products (AERONET, 2021) in the case of the long-term observations in the Amazon region, southern Africa, and Southeast Asia, and level-1.5 data in the case of the remaining stations. The reason for using level-1.5 data was to significantly increase the number of available observations in our smoke-related studies. Many observations showing high to very high smoke AOTs could not pass the strict criteria of the AERONET data quality checks and were thus removed from the level-2.0 data set. We compared the level 2.0 AERONET products with the corresponding
(reduced) level-1.5 products to guarantee that the used level-1.5 data set was of high quality.

In agreement with the AERONET data analysis of Sayer et al. (2014), we used the fine-mode AOTs stored in the AERONET data base. Smoke particles form a well developed accumulation mode (with sizes up to about 1 $\mu$m in radius) and the related optical properties are assigned as fine-mode AERONET products (Sayer et al., 2014). However, as will be discussed in Sect. 5.1, a bimodal distribution (accumulation plus coarse mode) was often retrieved from the AERONET sun and sky observations.
This was also pointed out by Sayer et al. (2014). The second mode is probably related to soil, road, and desert dust or marine aerosol in the planetary boundary layer. The comparison with respective lidar observations clearly indicate that smoke produces a pronounced accumulation mode, only. A coarse mode is absent. Thus, we computed the smoke-related values of $s$, $v$, $n_{50}$, and $n_{250}$ from the downloaded size distributions by considering the size classes 1-11 only (covering the accumulation mode and thus the radius range up to 0.9-0.95 $\mu$m), and correlated these calculated microphysical values with the fine-mode AOT at
532 nm as stored in the AERONET data base to finally obtain the conversion parameters. Details to the computation of $s$, $v$, $n_{50}$, and $n_{250}$ from the AERONET size distributions can be found in Mamouri and Ansmann (2016, 2017).

We begin the discussion of the AERONET results with an overview of the smoke measurements at Yellowknife and Churchill (stratospheric smoke), Reno and Table Mountain (tropospheric smoke), and at Punta Arenas, Rio Gallegos, and Marambio (stratospheric smoke) in Fig. 3. The downloaded AOT data sets (AERONET, 2021) contain values of fine-mode, coarse-mode,
and total AOT for 440, 675, 870 and 1020 nm. The AOT $\tau$ for 532 nm is obtained from the 440 nm AOT $\tau_{440}$ and the Ångström exponent $a$ by

$$\tau = \tau_{440}(440/532)^a. \tag{16}$$

The Ångström exponent $a$ is defined as $a = \ln(\tau_{440}/\tau_{675})/\ln(675/440)$ with wavelengths $\lambda$ of 440 and 675 nm. We separately computed 532 nm AOT for fine-mode, coarse-mode and total aerosol size distributions by using respective fine, coarse, and
total aerosol Ångström exponents. In Fig. 3, the total, i.e., fine-mode plus coarse-mode AOT is shown. In all other figures below, we exclusively used the fine-mode AOT at 532 nm. In cases with a strong smoke occurrence, the fine-mode fraction is usually >0.9.

The measurements at Yellowknife and Churchill in Fig. 3a were performed 0.5–2.5 days and 2–5 days after injection of smoke into the UTLS height range over British Columbia, Canada, respectively. The injection took place between 21:00 UTC on 12 August 2017 and 00:30 UTC on 13 August 2017 (Peterson et al., 2018). As can be seen, first smoke plumes arrived over Yellowknife, Canada, already 12-18 hours after injection. The 532 nm AOT reached values of almost 2.5. The smoke plumes travelled southeastward and crossed Churchill about 1.5-4 days later. A maximum AOT of 2.7 was measured over Churchill. At clean background conditions the AOT is about 0.025 to 0.05 at these Canadian AERONET stations. To consider all smoke observations over Yellowknife from 13-15 August 2017 (days 225-227) we set the AOT threshold level to 0.45, i.e., we considered cases with total 532 nm AOT of $\geq 0.45$, only, in our conversion study.

Rather strong fires occurred in California during the late summer and early autumn of 2020 (Fig. 3b). Mixtures of fresh and aged smoke were observed over Reno and Table Mountain. We increased the 532 nm total AOT threshold level to 0.6 to avoid a significant impact of urban haze on the wildfire smoke observations and derivation of smoke conversion parameters. The haze-related AOT was about 0.1-0.25. The exclusive use of the AERONET fine mode products further eliminated a potential impact of non-smoke aerosol such as coarse dust and marine particles on the correlation studies.

Figure 3c shows the observations of aged Australian wildfire smoke in southern South America and northern Antarctica. The smoke travelled more than 10000 km within 8-12 days before reaching our combined lidar and AERONET station at Punta Arenas (Ohneiser et al., 2020). The diluted smoke caused 532 nm AOTs mostly between 0.05 and 0.3. Maximum values were close to 0.5. At clean background conditions, the AOT is in the range from 0.025–0.035. In our smoke-related AERONET data analysis, we considered all observations with AOT>0.05 and again carefully checked that all used cases, even those with low AOT, showed clear and dominating smoke signatures (i.e., a pronounced accumulation mode). We selected the low AOT threshold of 0.05 to have sufficient cases in our conversion study for well-defined aged smoke. For each of the shown AOT observation in Figure 3 we downloaded the required size distributions and computed the respective column-integrated values of $s_{\mathrm{col}}$, $v_{\mathrm{col}}$, $n_{50,\mathrm{col}}$, and $n_{250,\mathrm{col}}$ (by considering the size classes 1–11).

To obtain the smoke extinction-to-volume conversion factor $c_{\mathrm{v}}$,

$$c_{\mathrm{v}} = \frac{v_{\mathrm{col}}}{\tau}, \tag{17}$$

required to derive volume and mass concentrations with Eqs. (1) and (5), the ratio of the vertically integrated (column) particle volume concentration $v_{\mathrm{col}}$ to the fine-mode 532 nm AOT $\tau$ was formed for each individual smoke observation. To facilitate the lidar-related discussion we divided the column values by an arbitary layer depth $D$ (length of the vertical column) and obtain

$$c_{\mathrm{v}} = \frac{v_{\mathrm{col}}/D}{\tau/D} = \frac{v}{\sigma} \tag{18}$$

with the layer mean volume concentration $v$ and the layer mean particle extinction coefficient $\sigma$. The introduced layer depth $D$ has no impact on the retrieval of the conversion factors and is only introduced to move from column-integrated values and AOT to more lidar-relevant quantities like concentrations and extinction coefficients. In this study, we set $D = 1000$ m as in the studies before (Mamouri and Ansmann, 2016, 2017).

For each smoke observation $j$ (from number $j = 1$ to $J$), available in the AERONET data base, we computed $c_{v,j}$ and then determined the mean value which we interpret as a representative smoke conversion factor,

$$c_v = \frac{1}{J} \sum_{j=1}^{J} \frac{v_j}{\sigma_j}. \tag{19}$$

In the same way, the conversion factors $c_{250}$, needed to estimate the large-particle number concentration with Eq. (3), and $c_s$, required in the surface-area retrieval with Eq. (2), were computed:

$$c_{250} = \frac{1}{J} \sum_{j=1}^{J} \frac{n_{250,j}}{\sigma_j}, \tag{20}$$

$$c_s = \frac{1}{J} \sum_{j=1}^{J} \frac{s_j}{\sigma_j}. \tag{21}$$

It is noteworthy to emphasize again that only the accumulation mode size range (radius classes 1-11) was considered in the computation of $n_{250}$ and $s$.

In the retrieval of the conversion parameters required to obtain $n_{50}$ (Eq. (4)), we used a different approach (Mamouri and Ansmann, 2016). Following the procedure suggested by Shinozuka et al. (2015), we applied a log-log regression analysis to the $\log(n_{50,j})$-$\log(\sigma_j)$ data field and determined in this way representative values for $c_{50}$ and $x$ that fulfill best the relationship,

$$\log(n_{50}) = \log(c_{50}) + x \log(\sigma). \tag{22}$$

## 5 AERONET results

We begin the result section with a discussion of observed smoke size distributions in Sect. 5.1. The continuous growth of smoke particles during the first days after emission is linked to a continuous change of the conversion factors. Therefore, the conversion parameters are significantly different for fresh and aged smoke. In Sect. 5.2, we then present the results of the AERONET-based correlation analysis, starting with the most simple scenarios of well-defined aged smoke observed over the AERONET stations in southern South America and northern Antarctica. Afterwards, we illuminate the link between the microphysical properties $v$, $s$, $n_{50}$ and $n_{250}$ and the measured light-extinction coefficient $\sigma$ for mixtures of fresh and aged smoke in North America (Sect. 5.3) and over the subtropical and tropical stations in South America, southern Africa, and Southeast Asia (Sect. 5.4). Supplementary, in Sect. 5.5, we compare the AERONET findings with lidar observations of smoke conversion factors. The lidar-based approach is an independent method to determine microphycial properties from measured optical effects and thus provides a favorably opportunity to check the relationship between microphysical and optical properties of smoke layers as obtained from the AERONET analysis.

## 5.1 Smoke particle size distributions: from fresh to aged smoke

As emphasized in Sect. 2, the particle size distribution of smoke particles changes with time during the first days after injection into the atmosphere as a result of particle aging processes (chemical processing, particle collisons and coagulation). The changing size distribution has a strong influence on the microphysical and optical properties and the correlation between $v$, $s$, $n_{50}$, and $n_{250}$ and the smoke extinction coefficient $\sigma$.

Figure 4 provides insight into the full range of size distributions of atmospheric smoke particles. The smallest particles found at Alta Floresta indicate rather fresh smoke, probably just a few hours after emission. The size distributions for Yellowknife (measured on 13 August 2017, 23:18 UTC) and Churchill were observed about 20 hours and 3.5 days after injection of smoke into the UTLS height region, respectively. Aged smoke after long-range transport over more than one week was observed at Punta Arenas (8 days after emission) and Lindenberg (10.5 days after emission). It is obvious that the size distribution is shifted towards larger particles with increasing residence time in the atmosphere. All size distributions are normalized so that the integral over each shown size distribution is one. Lidar observation conducted at Leipzig, 180 km to the southwest of Lindenberg (Haarig et al., 2018) and over Punta Arenas (Ohneiser et al., 2020) agree well with the respective AERONET size distributions. The lidar observations corroborate that the smoke size distribution is unimodal.

Figure 5 shows unimodal as well as bimodal size distribution in cases clearly dominated by smoke. Similar bimodal size distributions were presented in the smoke study of Sayer et al. (2014). The weak coarse mode may result from aerosols in the boundary layer (marine particles, soil and road dust). The lidar observations do not show this coarse mode.

To consider the changing smoke size distributions shown in Fig. 4 in the smoke data analysis it would be desirable to have conversion parameter sets for fresh, weakly aged, and aged smoke particles. However, in all likelihood such an approach would be impractical and /or unreasonably difficult. As will be discussed below in detail, the majority of AERONET smoke observations close to the fire regions indicates that fresh smoke was usually mixed with enhanced levels of background aerosol which, to a large extent, consists of aged smoke. This regional background aerosol obviously builds up over the fire regions during the long-lasting fire seasons. Therefore, we decided to distinguish just between two different measurement scenarios: (a) aged smoke observations (smoke observed after long range transport over 5 days and more) and (b) measurements of mixtures of fresh and aged smoke (in the near-range to large fire areas). For these two scenarios we developed conversion parameterizations.

## 5.2 AERONET results for aged smoke

Figure 6 shows the relationship between (a) the smoke volume concentration $v$ and the smoke-related extinction coefficient $\sigma$, (b) particle surface area concentration $s$ vs $\sigma$, and (c) the particle number concentration of larger smoke particles $n_{250}$ and $\sigma$ for aged Australian smoke. The correlation between the number concentration $n_{50}$ and $\sigma$ is discussed in Sect. 5.6. As a general impression, a clear relationship between $v$, $s$, and $n_{250}$ and $\sigma$ is found, at least up to extinction coefficients of $300\,\mathrm{Mm^{-1}}$ (or 0.3 in terms of the fine-mode AOT at 532 nm). The spread in the data reflects variations in the smoke properties (size distribution, refractive index). However, the relatively low scatter in the data is a sign for large similarities in the smoke properties (observed

over several weeks). This may be related to the fact that the flaming-fire type prevailed, eucalyptus trees were the main burning material, smoke lifting was always linked to strong pyroCb activity and thus similar lifting features, and the size distributions of aged smoke particles after 8-12 days long-range transport are at all very similar.

The mean relationship between $v$, $s$, and $n_{250}$ and $\sigma$ are visualized by straight blue lines. The respective mean conversion factors $c_v$, $c_s$, and $c_{250}$ are given as numbers in the different panels and also summarized in Table 3. These mean conversion factors were computed from the data in Fig. 6a, 6b, and 6c by using the Eqs. (19), (21), and (20), respectively.

### 5.3   AERONET results for mixtures of fresh and aged North American smoke

Figure 7 presents the correlations between the smoke volume concentration $v$ and the smoke extinction coefficient $\sigma$ (Fig. 7a) and between the smoke surface area concentration $s$ and the smoke extinction coefficient (Fig. 7a) for North American forest fires. To keep the discussions short, we concentrate on the most important retrievals only. The forests in the western United States and Canada mainly consist of pine, fir, aspen, and cedar trees. The flaming-fire type probably prevailed in August 2017 and August-October 2020. The observations in Fig.7 cover fresh and aged smoke plumes as well as mixtures of both. Strong variations in the size distribution are reflected in the comparably large scatter in the data. The upper part of the data fields show cases dominated by fresh smoke (smaller particles) and the lower part, around the blue regression line for aged smoke (from Fig. 6) is dominated by aged smoke (larger particles). Nevertheless, a clear relationship between the computed volume and surface area concentrations and the measured smoke extinction coefficient is given.

We used the observations at Yellowknife (1-2-days-old stratospheric smoke) and Reno (tropospheric smoke, observed a few hours to several days after injection) to compute the conversion parameters and mean relationships visualized by red solid lines in Fig. 7. The mean value of $c_v$ and $c_s$, as given in the figures, were calculated with Eqs. (19) and (21). Only the Yellowknife and Reno data in Fig. 7 were considered in this computation. All mean conversion factors are summarized in Table 3.

The Yellowknife data points (fresh smoke) are close the red lines. This may indicate that the respective conversion factors (given as numbers in Fig.7) describe predominately fresh and weakly aged North American smoke properties. The blue straight lines (for aged Australian smoke) seem to define the lower limit of the range of values in Fig.7. Many observations taken at Table Mountain, east of Los Angeles (tropospheric smoke) and at Churchill (2-5-days-old stratospheric smoke) are close to the blue lines for aged smoke.

### 5.4   AERONET results for mixtures of fresh and aged Amazonian, African, and Southeast Asian smoke

In this section, we switch from short-term observations of record-breaking and major fire episodes to long-term observations (partly over decades) in key burning areas of global importance. We assume that these long-term observations cover the full range of smoke-property-influencing aspects (smoldering and flaming fires, very different fuel types, short to large-range smoke transport and related smoke aging effects). Figure 8 presents the correlations between the computed smoke values of the volume concentration $v$ and surface area concentration $s$ and the smoke-related fine-mode extinction coefficient $\sigma$ at 532 nm for all four selected subtropical and tropical stations. A relatively strong variability is found for the relationship between the surface area concentration and extinction coefficient in Fig. 8b, even significant differences between the different data sets (Southeast

Asian vs African and Amazonian observations) are visible. In contrast, a quite narrow distribution of all observations is given for the volume-to-extinction relationship in Fig. 8a.

The spread in the data is again widely a function of the size distribution and thus of the age of the smoke layers. As in Fig. 7, the upper part of the data fields is strongly influenced by smaller particles and thus fresh smoke, whereas the lower part is controlled by larger particles and thus aged smoke.

The green straight lines show the mean regression lines for the Mongu-Zambia data set. The computation of the mean conversion factors is performed in the same way as described in the forgoing sections. We included again the mean regression lines for aged Australian smoke (blue lines) and also for comparably fresh North America smoke (red lines) in Fig. 8. It is obvious that the blue lines for aged smoke indicate well the lower boundary of the data range in Fig. 8a and b. On the other hand, the upper boundary of the data field seems to be less well defined. Obviously many of the observed plumes of tropical and subtropical fires, especially over Zambia and the Amazon region, are just a few hours old and thus the smoke particles were very small. The smoke particles of the Amazon region, southern Africa, and Southeast Asia, are frequently considerably smaller than North American smoke particles (represented by the red lines in Fig. 8).

It is noteworthy to mention that Sayer et al. (2014) analyzed the relationship between the colum smoke volume concentrations $v_{col}$ and the 550 nm fine-mode AOT $\tau_{550}$ for a large number of AERONET stations around the world with strong impact of wild fires smoke and found similar mean values for the ratio $v_{col}/\tau_{550}$ as given for the extinction-to-volume conversion factor $c_v$ in our figures and in the summarizing Table 3. The study of Sayer et al. (2014) includes also Russian stations (Moscow, Tomsk, Yakutsk). We may thus conclude that our conversion parameter set well covers main aspects and characteristics of wildfire smoke layers around the world.

## 5.5 AERONET vs lidar smoke observations

Lidar provides an independent approach to derive microphysical parameters of smoke and thus to determine the link between the retrieved microphysical and measured optical properties of smoke particles (Müller et al., 1999a, 2014; Veselovskii et al., 2002, 2012). This option provides the favorable opportunity to check the quality and robustness of our results obtained by analyzing the AERONET data. One of the main problems of sun photometer observations is that the entire vertical column is observed so that, e.g., boundary-layer aerosols can be a disturbing factor in the study of lofted tropospheric and stratospheric smoke plumes. These problems are absent in the case of profiling techniques such as lidar. In the case of active remote sensing methods, the optical and microphysical properties are exclusively determined for the smoke layers. However, the uncertainties in the lidar retrievals can be large and thus the obtained data products can scatter over a wide range just as function of these uncertainties.

In Fig. 9, lidar data sets of smoke observations from 53°S (Punta Arenas) to 86°N (North Pole range) are considered. Correlation between $v$ and $s$ values and $\sigma$ for fresh and aged smoke plumes originating from fires in western Canada, eastern Siberia, southeastern Australia, eastern United States, the Amazon Basin, and central western Africa are shown. The AERONET-derived mean relationship between $v$, $s$, and $n_{250}$ and $\sigma$ for aged, fresh and the long-term African observations as discussed in the foregoing sections are shown again as blue, red, and green lines.

A large scatter in the lidar-based smoke correlation values is visible in Fig. 9 with data points even below the blue lines and above the green lines. This large scatter is partly related to the specific retrieval methodology and data analysis strategy as well as to varying assumptions in the analysis of the different lidar data packages. The most robust results (less sensitive to input errors) are obtained in terms of surface area concentrations when using the inversion algorithm of Müller et al. (1999a, b). This method was applied to the lidar observations at Praia, Cabo Verde, Manaus, Brazil, and Lindenberg, Germany. The other observations taken at Leipzig, Punta Arenas, and the North Pole region were analyzed by applying the analysis scheme of Veselovskii et al. (2002).

In Fig. 9b, it can be seen that most of the smoke layers observed over Praia (smoke from central western Africa) contain aged smoke particles (the data points are close to the blue line), and only a minor part of the observations indicate fresh smoke plumes (these data points are close to the green line). Many smoke layers contained a mixture of fresh and aged smoke. All the lidar data, representing smoke after long range transport (Lindenberg, Leipzig, North Pole, Punta Arenas) are close to the blue line for aged smoke or even below this line, and thus in good agreement with the AERONET-based correlation studies. From the found consistency in the shown correlations, based on AERONET and lidar observations, we can conclude that the AERONET smoke conversion parameters presented here allowa trustworthy retrieval of smoke microphysical properties from backscatter lidar observations.

## 5.6 AERONET results: $n_{50}$ retrieval

Figure 10 shows the correlation between the CCN-relevant particle number concentration $n_{50}$ and the extinction coefficient $\sigma$ for two contrasting smoke data sets, i.e., for the observations of aged Australian smoke and, on the other hand side, for the observations of fresh smoke (Yellowstone) and mixtures of fresh and aged smoke (Reno). According to the applied regression analysis, fresh smoke plumes contain much more CCN-relevant small particles (roughly a factor of 3 more) than aged plumes. For a given extinction coefficient of $\sigma$=100 Mm$^{-1}$, $n_{50}$ is 635 cm$^{-3}$ (for aged Australian smoke over Punta Arenas), 1900 cm$^{-3}$ (for North American smoke), and 3200 cm$^{-3}$ (for Mongu, Zambia). The numbers for $n_{50}$ and the extinction exponent $x$ (see Eq. 4) in Fig. 10 and Table 3 are obtained by considering the respect data sets shown in the figure or mentioned in the Table in the linear regression analysis described in Sect. 4.2.

## 6 Summary of AERONET-derived conversion parameters and retrieval uncertainties

Table 3 provides an overview of the derived mean conversion parameters for the different AERONET observational data sets, discussed in the foregoing section. Since the smoke size distribution widely controls the derived conversion parameters, we added the information on the effective radius, which is the particle-surface-area-weighted mean radius of the smoke accumulation mode and can be regarded as a typical radius of the observed smoke particles. For aged smoke, the effective radius is largest. It is much lower for the mixtures of fresh and aged smoke. We recommended to use the two conversion parameter sets in the lower part of Table 3 in the analysis of smoke layers observed with backscatter lidars.

In Table 4, the uncertainties in the input parameters and the smoke retrieval products are listed. The uncertainties in the conversion parameters are estimated from the SD values in Table 3. The relative uncertainties in the required smoke lidar ratio $L$ and smoke particle density $\rho$ follow from the discussions in Sect. 3. Three scenarios of lidar backscatter profiling are compared in Table 4. In the case of a Raman lidar or a HSRL, the determination of the particle backscatter coefficient in clearly identified smoke layers is possible with high accuracy (10% relative uncertainty) as our experience shows (Wandinger et al., 2002; Veselovskii et al., 2015; Haarig et al., 2018; Ohneiser et al., 2020; Ohneiser et al., 2021). In addition, the lidar ratio $L$ is measured with a typical relative uncertainty of around 20%. In the case of a powerful ground-based elastic-backscatter lidar, the smoke lidar ratio must be estimated in the determination of the extinction coefficient. The lidar ratio is even required as input in the basic determination of the backscatter coefficient profiles. The backscatter profile retrieval may be possible with a relative uncertainty of 15%. In the case of comparably weak backscatter signals measured from space (e.g., with the CALIPSO lidar), we assume an uncertainty of 25% in Table 4 in the profiling of the backscatter coefficient. Details to the uncertainties in the CALIPSO aerosol backscatter coefficients are given in Young (2013, 2018). Finally, the relative uncertainties in the smoke microphysical retrieval products are obtained by error propagation applied to Eqs. (1)-(5) in Sect. 3.

As can be seen in Table 4, the retrieval of volume, mass, and surface-area concentrations of detected smoke layers is possible with an overall uncertainty of about 25-35% (for both fresh or near-source smoke and for aged smoke after long range transport) in the case of Raman lidars or HSRLs, when the smoke lidar ratios are measured. The respective uncertainties are 40-50% when smoke profiling is performed with an elastic backscatter lidar so that the lidar ratio $L$ needs to be estimated. The number concentrations $n_{50}$ and $n_{250}$ can be only roughly estimated with a typical uncertainty of about 50-70%. Again, the retrieval uncertainties are lowest when measurements are performed with a ground-based Raman lidar or a HSRL. The uncertainties are then of the order of 35-50% in the case of aged smoke.

Uncertainties in the estimates of CCN and INP concentrations are not listed in Table 4. Comparison with airborne in situ observations of CCN profiles suggest that the uncertainty in the lidar-based CCN estimation is around 50%, and in extreme cases up to a factor of 2 (Düsing et al., 2018; Haarig et al., 2019; Genz et al., 2020). In the case of INP estimations, it is too early for an in-depth uncertainty analysis. A considerable number of dedicated field campaigns and further laboratory studies are needed before a trustworthy quantification of uncertainties in the INP estimation is possible (see also the discussion at the end of Sect. 3.1).

At the end of the section it should be mentioned that the developed method (here for 532 nm) can be applied to single-wavelength 355 nm and 1064 nm backscatter lidar observations as well. We recommend in these cases to estimate the 532 nm backscatter profiles from the measured 355 or 1064 nm backscatter profiles by using properly estimated smoke color ratios $\beta_{532}/\beta_{355}$ and $\beta_{532}/\beta_{1064}$ (the index denotes wavelength in nm). Extended overviews of observed wavelength dependencies of smoke backscatter coefficients can be found in Burton et al. (2012) and Adam et al. (2020). In a follow-on project, we may repeat the procedure presented here for 532 nm for the wavelength of 355 nm to cover spaceborne 355 nm HSRL lidar observations of the European Space Agency. Such an approach was already presented by Mamouri and Ansmann (2016, 2017) in the case of the marine and Saharan dust types.

## 7 Lidar case studies

We applied the new smoke conversion scheme to two contrasting smoke observations. In Fig. 1, an aged stratospheric Australian smoke layer was shown observed with an advanced multiwavelength Raman lidar (Polly: POrtabLe Lidar sYstem) (Engelmann et al., 2016) at Punta Arenas, Chile, in January 2020. This case will be further analyzed in Sect. 7.1. As a second contrasting example, we selected a measurement of the spaceborne CALIPSO lidar over North and South Dakokota, USA. A comparably fresh tropospheric smoke layer was detected in September 2020. The smoke originated from strong wild fires in the western part of the United States and Canada. This case study is presented in Sect. 7.2.

### 7.1 Aged Australian smoke in the stratosphere observed with ground-based Raman lidar

In the framework of a multiyear measurement campaign, we monitored the stratospheric perturbation caused by the record-breaking Australian bushfires with a polarization Raman lidar Polly over a full year, starting in January 2020 (Ohneiser et al., 2020). A measurement example is shown in Fig. 1. The results obtained by applying the conversions scheme in Sect. 3 are presented in Fig. 11–13. In the first step, we calculated the extinction coefficients from the 532 nm backscatter coefficients by using a smoke lidar ratio of $L$=95 sr as measured with the Raman lidar Polly (Ohneiser et al., 2020). Then we applied the conversion factor $c_v$ in Table 3 for aged smoke to obtain the volume concentration $v$. By assuming a particle density of $\rho = 1.15$ g cm$^{-3}$ for the smoke particles, we obtain the mass concentration $m$ shown in Fig. 11.

Such a high aerosol pollution level of 15 $\mu$g m$^{-3}$ at heights from 20-26 km height has never been observed in the stratosphere before, even not after major volcanic eruptions (Trickl et al., 2013; Sakai et al., 2016). Stratospheric background levels are of the order of 0.01 $\mu$g m$^{-3}$ (Baars et al., 2019; Taha et al., 2020). As shown in Fig. 1, the particle depolarization ratio was significantly enhanced as a result of fast lifting by pyroCb clouds over the Australian fire regions (Ohneiser et al., 2020). The aging process was obviously not fully completed and the particles were probably glassy. This may explain the deviation from the perfect spherical shape of the particles and the enhanced depolarization ratios (Gialitaki et al., 2020).

Figure 12 shows the derived surface area concentration $s$ and the particle number concentration $n_{50}$. Information on number concentrations and surface area at stratospheric heights is of interest, e.g., in PSC and ozone research. A record-breaking ozone depletion was observed in the stratosphere over Antarctica starting in September 2020 (CAMS, 2021). PSC particles play a strong role in this context because they permit the activation of chlorine components (on the surfaces of the PSC particles) which subsequently destroy ozone molecules. Even if we assume a strong decay of the stratospheric smoke perturbation by a factor of 10 or 100 in the southern hemisphere (at mid to high latitudes) from January 2020 to September 2020, and thus a reduction in the smoke number concentration from about 500 cm$^{-3}$ in Fig. 12 (in the height range from 21 to 25.5 km height) to 50 or even 5 cm$^{-3}$, such number concentrations are still high and are in the range of particle concentrations typically observed in PSCs (0.01–10 cm$^{-3}$) (Jumelet et al., 2008, 2009). Smoke particles may be able to serve as nuclei in processes of heterogeneous nucleation of PSC particles (Hoyle et al., 2013; Zhu et al., 2018) and thus may influence PSC microphyscial properties. On the other hand, smoke surface area concentrations of around 120-130 $\mu$m$^2$ cm$^{-3}$ in the stratospheric layer in Fig. 12 are extremely high, and even if the smoke concentration was reduced by a factor of 10 to 100 until September 2020,

surface area concentrations of around 10 or around 1 $\mu$m$^2$ cm$^{-3}$ are still high and partly in the same range of typical surface area concentrations in PSC clouds (Jumelet et al., 2008, 2009) so that at least a weak influence on ozone depletion by providing surface areas for chlorine activation can not be excluded.

The surface area concentration is also an essential aerosol input parameter in the INP parameterization and thus an important quantity in the research field of aerosol-cloud interaction with focus on mixed-phase-cloud and cirrus formation in the troposphere. INP estimates are shown in Fig. 13. We use the aerosol type parameters for Leonardite as given in Table 2 in the calculation of immersion freezing INP ($n_{\mathrm{INP,I}}$, Sect. 3.1.1). The calculations start with the computation of the water activation criterion $\Delta a_{\mathrm{w}}$ (Eq. 8). Ice nucleation is a strong function of the vertical velocity (lifting of moist air parcels) which leads to ice supersaturation and thus determines $\Delta a_{\mathrm{w}}$. In the case study here, we assume realistic upper-tropospheric cirrus formation conditions and ignore in this demonstration of INP number estimation that we observed the smoke layer in the dry stratosphere 10-15 km above the local tropopause. We assumed $RH_{\mathrm{w}} = 79.85\%$ and $82.35\%$ and a temperature $T$ of $-50°$C. The corresponding $RH_{\mathrm{i}}$ values are around 125% and 130%. Homogeneous freezing proceeds in significant numbers at about $RH_{\mathrm{i}} = 150\%$ at $-50°$C. Thus, for slow air lifting, smoke particles potentially acting as INPs have a good chance to sensitively influence cirrus formation. With these input values for $RH_{\mathrm{w}}$ and $T$, we obtain $\Delta a_{\mathrm{w}} = 0.175$ and $0.2$. The value for the ice melting point $a_{\mathrm{w,i}}$ (Eq. 9) is 0.6235 at $-50°$C. Afterwards, we calculated the ice nucleation rate $J_{\mathrm{het,I}}$ (Eq. 10) and the INP concentration $n_{\mathrm{INP,I}}$ (Eq. 11) by assuming a lifting period of 600 s during which ice supersaturation conditions according to $\Delta a_{\mathrm{w}}$ of 0.175 and 0.2 are given. We also computed deposition nucleation INP solutions ($n_{\mathrm{INP,D}}$, Sect. 3.1.3) by assuming the same $T$, $RH_{\mathrm{i}}$, $RH_{\mathrm{w}}$, and $S_{\mathrm{i}}$ input parameters together with an overall lifting period of 600 s.

Figure 13 shows the results of the $n_{\mathrm{INP,I}}$ and $n_{\mathrm{INP,D}}$ estimations. A strong dependence on relative humidity and ice supersaturation is visible. Obviously a threshold value of ice supersaturation $S_{\mathrm{i}}$ has to be reached and exceeded before efficient immersion freezing in the case of Leonardite starts. The estimated deposition nucleation INP concentration is much higher than the immersion-freezing INP values for the assumed atmospheric conditions. The obtained high INP numbers are directly correlated to the large amount of smoke particles. The obtained INP number concentrations are not too uncommon. For example, INP number concentrations reached about 10-100 L$^{-1}$ in a Saharan dust plume (DeMott et al., 2003). Neglecting any radiative heating effects of the smoke layer and microphysical processes such as sedimentation and competition for water vapor, these results clearly indicate that organic smoke particles can impact ice formation processes in the upper troposphere during favorable moisture conditions and gravity wave activity with updraft phases lasting longer than several minutes.

In Fig. 13, also values for $n_{250}$ (large particle fraction) are shown. It is usually assumed that particles with diameters >500 nm can be regarded as the overall reservoir for INPs (DeMott et al., 2015). Number concentrations of 10-100 cm$^{-3}$ or 10000-100000 L$^{-1}$ indicate that this reservoir of large smoke particles cannot be depleted when $n_{\mathrm{INP}}$ is in the range of 0.1 to 100 L$^{-1}$, even not during extended cirrus formation processes lasting several hours.

The competitive process to heterogeneous ice nucleation is homogeneous freezing. If ice supersaturation $S_{\mathrm{i}}$ reaches sufficient levels, corresponding to $\Delta a_{\mathrm{w}}$ of 0.29-0.31, $n_{\mathrm{INP,hom}}$ (Eq. 13) would be of the order of 600-700 L$^{-1}$ for $v \approx 10^{-8}$ cm$^3$ L$^{-1}$ (mean value of the 20-26 km layer).

As mentioned the uncertainty in the INP retrieval is large and is widely related to the current status of our knowledge about smoke INP type characteristics. The lidar input parameters $s$ and $v$ could be obtained with low relative errors of 25-35%. The research on the role of wildfire smoke particles in cirrus and PSC formation is one of the key topics in atmospheric research with focus on aerosol-cloud interaction (Knopf et al., 2018).

## 7.2 North American smoke in the troposphere observed with the CALIPSO lidar

Strong fires occurred in the western United States and western Canada during the late summer of 2020. The smoke reached even Europe (Baars et al., 2021). Figure 14 shows an overflight of CALIPSO from North Dakota down to Texas. Two smoke layer complexes were detected between 5 and 10 km height, one over North and South Dakota and another one over Texas. According to the backward trajectory analysis, the plumes over North and South Dakota originated from fires in western Canada and were observed after one day of travel from the main fire area to North and South Dakota. The plumes over Texas originated from heavy Californian fires and were observed after 3-5 days of travel time from the Californian smoke sources. Cirrus formed in the neighborhood of the smoke layers.

Figure 15 presents the height profiles of smoke extinction coefficient, mass concentration, surface area concentration and estimated INP concentration for the smoke layers detected over eastern North and South Dakota. We used a lidar ratio of 70 sr to convert the measured smoke backscatter coefficients into extinction values and applied the conversion parameter set for fresh smoke as recommended in Table 3. A potential influence of multiple scattering was ignored. For dense aerosol layers, multiple scattering can introduce substantial unquantified errors into the CALIPSO lidar retrievals of particle backscatter and extinction coefficients (Wandinger et al., 2010; Liu et al., 2011). However, multiple scattering effects may be of the order of about 5-10% (underestimation of the true extinction coefficient by 5-10%) in cases with smoke layer optical thickness $\leq 0.5$ at 532 nm (Prata et al., 2017),. The particle depolarization ratio was >0.1 and thus indicated the presence of nonspherical smoke particles.

According to Table 4 (third column), the uncertainties in the lidar products are higher now compared to measurements with ground-based Raman lidar at Punta Arenas. Relative uncertainties of 40-45% in the extinction coefficient are mainly caused by the lidar ratio assumption. The uncertainties in the mass and surface area concentrations are around 50%.

In the computation of the immersion-freezing INP concentrations in Fig. 15b, we highlight the impact of the selected organic aerosol type (Leonardite, LEO, vs free tropospheric aerosol, FTA, Table 2). We assumed a water activity criterion of $\Delta a_{\mathrm{w}} = 0.2$ or $RH_{\mathrm{i}}$=130% at $T = -40°C$, and again set the period during which ice nucleation was possible at these thermodynamic conditions to $\Delta t = 600$ s. As can be seen, the assumed organic substance can have a very sensitive impact on the estimated INP values. The third organic substance in Table 2 (Pahokee Peat) leads to similar INP values as obtained for Leonardite. In cirrus research, it is thus essential to know the origin of the smoke and a good knowledge of the burning material to be able to properly characterize the aerosol type involved in the cloud formation studies.

## 8 Conclusion/Outlook

We presented a new method that permits the retrieval of tropospheric and stratospheric height profiles of smoke particle mass, volume, surface area, and number concentrations as well as first-order estimates of CCN and INP concentrations from single-wavelength backscatter lidar observations. The developed smoke retrieval method is of special importance for spaceborne backscatter lidars such as CALIPSO and the numerous ground-based lidars permitting high-quality observations of height profiles of the particle backscatter coefficient at 532 nm up to stratospheric heights. The method allows us to characterize smoke microphysical and optical properties even at very low smoke pollution levels and thus during the entire decay phase of long-lasting stratospheric perturbations, from thick smoke plumes to aerosol background conditions. Even if advanced multiwavelength Raman or HSR lidar observations are available for the characterization of pronounced smoke layers so that the lidar inversion procedure can be applied to obtain the microphysical properties, our method based on conversion factors is useful for comparisons to corroborate the quality of the solutions obtained with advanced multiwavelength lidar systems.

The required conversion factors were determined from AERONET observations. In this approach, we distinguished observations of aged smoke and and mixtures of fresh and aged smoke. A crucial task is the estimation of smoke INP concentrations because of the complex characteristics of smoke particles. Now, a consistent methodology is available to characterize wildfire smoke plumes in terms of microphysical and cloud-relevant parameters. This will allow us to study smoke-cirrus interaction as well as a potential impact of smoke particles on PSC formation and ozone depletion in large detail. We applied the new smoke analysis scheme to ground-based as well as spaceborne CALIPSO observations to highlight the potential of single-wavelength lidars (at ground and in space) to significantly contribute to an extended monitoring and microphysical characterization of tropospheric and stratospheric smoke layers and thus to provide valuable information for climate-, cloud-, and air chemistry modeling efforts.

## 9 Data availability

Polly lidar observations (level 0 data, measured signals) are in the PollyNET data base (Pollynet, 2021) with quicklooks at http://polly.tropos.de. All the analysis products are available at TROPOS upon request (info@tropos.de). CALIPSO observations of smoke profiles and smoke AOT were used and downloaded from the CALIPSO data base (CALIPSO, 2020a, b, c). AERONET observations were downloaded from the AERONET data base (AERONET, 2021).

## 10 Author contributions

The paper was written by AA with contributions (data analysis, methodological concepts) from KO, REM, DAK, IV, HB, and AF. The co-authors RE, CJ, PS, and BB were involved in the field observations and took care of all the smoke measurements with the Polly lidars at Punta Arenas and aboard RV Polarstern.

## 11  Competing interests

The authors declare that they have no conflict of interest.

## 12  Financial support

The authors acknowledge support through the European Research Infrastructure for the observation of Aerosol, Clouds and

Trace Gases ACTRIS under grant agreement no. 654109 and 739530 from the European Union's Horizon 2020 research and
innovation programme. We thank AERONET-Europe for providing an excellent calibration service. AERONET-Europe is part
of the ACTRIS project. R.-E. M. has been financial supported by the SIROCCO project (grant no. EXCELLENCE/1216/0217)
co-funded by the Republic of Cyprus and the structural funds of the European Union for Cyprus through the Research and
Innovation Foundation. Thanks are also provided to the ERATOSTHENES Centre of Excellence which was established af-

ter receiving funding by the Republic of Cyprus and the EU H2020 Widespread Teaming program with Grant Agreement
No 857510 (www.excelsior2020.eu). The field observations at Punta Arenas are partly funded by the German Science Foun-
dation (DFG) project PICNICC with project number 408008112. The development of the lidar inversion algorithm used to
analyze Polly data was supported by the Russian Science Foundation (project 16-17-10241). D.K. acknowledges support by
the DOE grant DE-SC0021034. The Polarstern Polly data was produced as part of the international Multidisciplinary drifting

Observatory for the Study of the Arctic Climate (MOSAiC) with the tag MOSAiC20192020 and Project ID AWI_PS122_00.

The publication of this article was funded by the Open Access Fund of the Leibniz Association

*Acknowledgements.* We thank AERONET for their continuous efforts in providing high-quality measurements and products. We are grateful
to the present AERONET site managers Jacobo Salvador, Raul D'Elia, Ramiro Gonzales, and Jonathan Ferrarae (Ceilap-RG, Marambio),
Norman O'Neill, Ihab Abboud, and Vitali Fioletov (Yellowknife, Churchill), Pam Glatfelter, Heath Rhoades, and William Buehlman (Table

Mountain California), W. Patrick Arnott and S. Marcela Loria-Salazar (Reno), Ralf Becker (Lindenberg), Edilson Bernadino de Andrade and
Fernando Morais (Alta Floresta), Mukufute Mukulabai (Mongu), Anthony Daka (Mongu-Inn), Surasak Meesiri, Anuson Niyompam, and
Anucha Yangthaisong (Mukdahan), and Tan Li (Singapore), but also to all previous site managers. We also thank the CALIPSO team for
their well-organized easy-to-use internet platforms.

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

**Table 1.** Dual-wavelength lidar oberservations of lidar ratios ($L$) at 355 and 532 nm in tropospheric (T) and stratospheric (S) smoke layers.

| Atmospheric layer | $L$(355 nm) | $L$(532 nm) | Reference |
|---|---|---|---|
| Aged Canadian smoke (S) | 35–50 sr | 50-80 sr | Haarig et al. (2018) |
| Aged Australian smoke (S) | 50-95 sr | 70–110 sr | Ohneiser et al. (2020) |
| Aged Canadian smoke (T) | 65 sr | 90 sr | Wandinger et al. (2002) |
| Aged Siberian smoke (T) | 40 sr | 65 sr | Murayama et al. (2004) |
| North American smoke (T) | 65–90 sr | 65–80 sr | Veselovskii et al. (2015) |
| European smoke (T) | 60–65 sr | 60–65 sr | Alados-Arboledas et al. (2011) |
| European smoke (T) | 30–60 sr | 45-65 sr | Nicolae et al. (2013) |
| European smoke (T) | 40–105 sr | 40-110 sr | Mylonaki et al. (2018) |
| Amazonian smoke (T) | 50–75 sr | 50–80 sr | Baars et al. (2012) |
| Western African smoke (T) | 50–110 sr | 50–105 sr | Tesche et al. (2011) |
| South African smoke (T) | 70–110 sr | 60–105 sr | Giannakaki et al. (2015) |

**Table 2.** Values for $b$ and $k$ for three organic-aerosol INP types required to determine the ice nucleation rate $J_{\mathrm{het,I}}$ with Eq. (10).

| INP type | $b$ | $k$ | Reference |
|---|---|---|---|
| Pahokee Peat (organic substance) | -15.78 | 78.31 | Knopf and Alpert (2013) |
| Leonardite (organic substance) | -13.40 | 66.90 | Knopf and Alpert (2013) |
| Free tropospheric aerosol (smoke plumes over Azores) | 0.656 | 2.981 | China et al. (2017) |

**Table 3.** Smoke conversion parameters required in the conversion of the particle extinction coefficient $\sigma$ at 532 nm into particle number concentrations $n_{50}$ and $n_{250}$, surface area concentration $s$, and volume concentration $v$. The mean values and SD for $c_v$, $c_s$, $c_{250}$, $c_{50}$, and $x$ are obtained from the extended AERONET data analysis. Effective radius $r_{\text{eff}}$ information is given as well. The conversion factors are derived from the AERONET observations at Yellowknife (Y), Reno (R), Alta Floresta (AF), Punta Arenas (PA), Rio Gallegos (RG), Maramabio (Ma), Mongu (Mo), Mukdahan (Mu), and Singapore (S). The conversion parameters for South America (AF), southern Africa (Mo), and Southeast Asia (Mu, S) consider observations with AOT>0.9 at 532 nm only.

| Observation (site) | $c_v$ $[10^{-12}\,\text{Mm}]$ | $c_s$ $[10^{-12}\,\text{Mm}\,\text{m}^2\,\text{cm}^{-3}]$ | $c_{250}$ $[\text{Mm}\,\text{cm}^{-3}]$ | $c_{50}$ $[\text{cm}^{-3}]$ | $x$ | $r_{\text{eff}}$ $[\text{nm}]$ |
|---|---|---|---|---|---|---|
| Aged smoke | | | | | | |
| S. Amer./Antarct. (PA, RG, Ma) | $0.129 \pm 0.009$ | $1.75 \pm 0.22$ | $0.354 \pm 0.081$ | $16.7 \pm 5.0$ | $0.79 \pm 0.08$ | $0.22 \pm 0.03$ |
| Fresh and mixtures of fresh and aged smoke | | | | | | |
| North America (Y, R) | $0.149 \pm 0.019$ | $2.67 \pm 0.52$ | $0.187 \pm 0.054$ | $50 \pm 15$ | $0.79 \pm 0.06$ | $0.17 \pm 0.01$ |
| South America (AF) | $0.163 \pm 0.018$ | $3.16 \pm 0.47$ | $0.151 \pm 0.045$ | $112 \pm 21$ | $0.73 \pm 0.02$ | $0.16 \pm 0.01$ |
| Southern Africa (Mo) | $0.162 \pm 0.020$ | $3.30 \pm 0.42$ | $0.113 \pm 0.021$ | $106 \pm 50$ | $0.74 \pm 0.09$ | $0.15 \pm 0.01$ |
| Southeast Asia (Mu, S) | $0.169 \pm 0.018$ | $2.68 \pm 0.47$ | $0.320 \pm 0.103$ | $111 \pm 80$ | $0.67 \pm 0.09$ | $0.18 \pm 0.03$ |
| Recommended smoke parameterization | | | | | | |
| Observations close to fire regions (fresh+aged smoke) | $0.16 \pm 0.02$ | $3.0 \pm 0.6$ | $0.18 \pm 0.09$ | $100 \pm 50$ | $0.75 \pm 0.08$ | |
| Observations far away from fire regions (aged smoke) | $0.13 \pm 0.01$ | $1.75 \pm 0.25$ | $0.35 \pm 0.08$ | $17 \pm 5$ | $0.79 \pm 0.08$ | |

**Table 4.** Relative uncertainties in the conversion input parameters (upper part of the table) and in the retrieved smoke products (lower part of the table). Fresh stands for mixtures of fresh and aged smoke (or for near-source smoke). Different lidar systems (Raman lidar/HSRL, ground-based elastic backscatter lidar, and spaceborne elastic backscatter lidar) and thus different uncertainties in the backscatter and lidar ratio profiles are considered. The uncertainties in the conversion factors and extinction exponents are estimated from Table 3. The smoke extinction coefficient is defined as $\sigma = L\beta$.

| | Raman lidar/HSRL | | Backscatter lidar (ground-based) | | Backscatter lidar (spaceborne) | |
|---|---|---|---|---|---|---|
| uncertainty | fresh | aged | fresh | aged | fresh | aged |
| $\delta\beta/\beta$ | 0.10 | 0.10 | 0.15 | 0.15 | 0.25 | 0.25 |
| $\delta L/L$ | 0.20 | 0.20 | 0.35 | 0.35 | 0.35 | 0.35 |
| $\delta c_{\mathrm{v}}/c_{\mathrm{v}}$ | 0.10 | 0.10 | 0.10 | 0.10 | 0.10 | 0.10 |
| $\delta c_{\mathrm{s}}/c_{\mathrm{s}}$ | 0.20 | 0.15 | 0.20 | 0.15 | 0.20 | 0.15 |
| $\delta c_{250}/c_{250}$ | 0.50 | 0.25 | 0.50 | 0.25 | 0.50 | 0.25 |
| $\delta c_{50}/c_{50}$ | 0.50 | 0.30 | 0.50 | 0.30 | 0.50 | 0.30 |
| $\delta x/x$ | 0.10 | 0.10 | 0.10 | 0.10 | 0.10 | 0.10 |
| $\delta\rho/\rho$ | 0.20 | 0.20 | 0.20 | 0.20 | 0.20 | 0.20 |
| $\delta\sigma/\sigma$ | 0.22 | 0.22 | 0.38 | 0.38 | 0.43 | 0.43 |
| $\delta v/v$ | 0.25 | 0.25 | 0.39 | 0.39 | 0.44 | 0.44 |
| $\delta m/m$ | 0.32 | 0.32 | 0.44 | 0.44 | 0.48 | 0.48 |
| $\delta s/s$ | 0.35 | 0.27 | 0.43 | 0.41 | 0.47 | 0.46 |
| $\delta n_{250}/n_{250}$ | 0.55 | 0.34 | 0.63 | 0.46 | 0.66 | 0.50 |
| $\delta n_{50}/n_{50}$ for $\sigma = 10\ \mathrm{Mm}^{-1}$ | 0.56 | 0.39 | 0.60 | 0.46 | 0.62 | 0.49 |
| $\delta n_{50}/n_{50}$ for $\sigma = 100\ \mathrm{Mm}^{-1}$ | 0.64 | 0.50 | 0.68 | 0.56 | 0.70 | 0.58 |

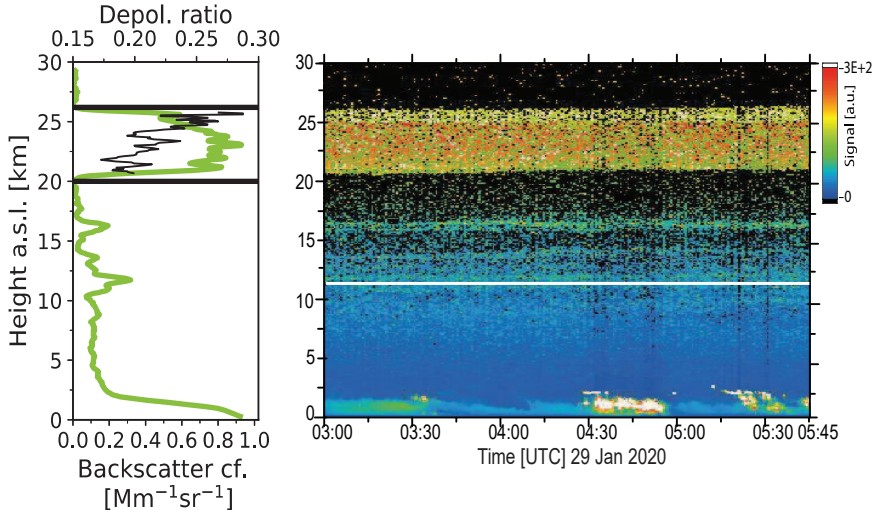

**Figure 1.** Australian bushfire smoke (yellow layer) in the stratosphere, almost 10-15 km above the tropopause (white line in the right panel). The mean backscatter coefficient profile (green) and the particle depolarization-ratio profile (black, for the main layer only) for the 165-minute observation are shown in the left panel. Main smoke layer base and top height are indicated by black horizontal lines in the left panel. The smoke was observed with lidar at Punta Arenas, Chile, on 29 January 2020, about 10000 km downwind of the Australian fire areas. The range-corrected 1064 nm lidar return signal is shown.

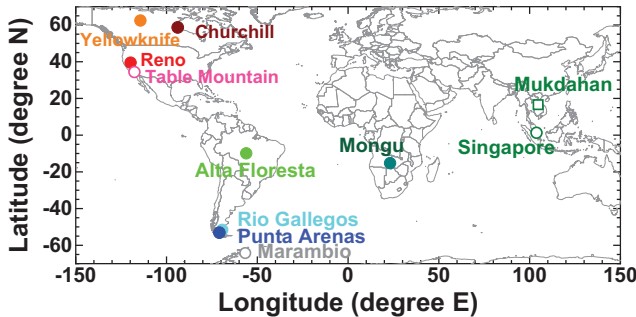

**Figure 2.** AERONET stations used in our study. Aged stratospheric smoke from the major Australian bush fires was observed over the South American and Antarctic stations (Rio Gallegos, Punta Arenas, Marambio) in January and February 2020. Fresh and aged stratospheric smoke from record-breaking fires in British Columbia, Canada, were measured over Yellowknife and Churchill, respectively, in August 2017. Mixtures of fresh and aged tropospheric smoke originating from strong fires in the western United States and Canada were found over Reno and Table Mountain in late August to mid October 2020. AERONET stations at Alta Floresta, Mongu, Mukdahan and Singapore have long, multiyear data records of smoke observations in key regions of biomass burning.

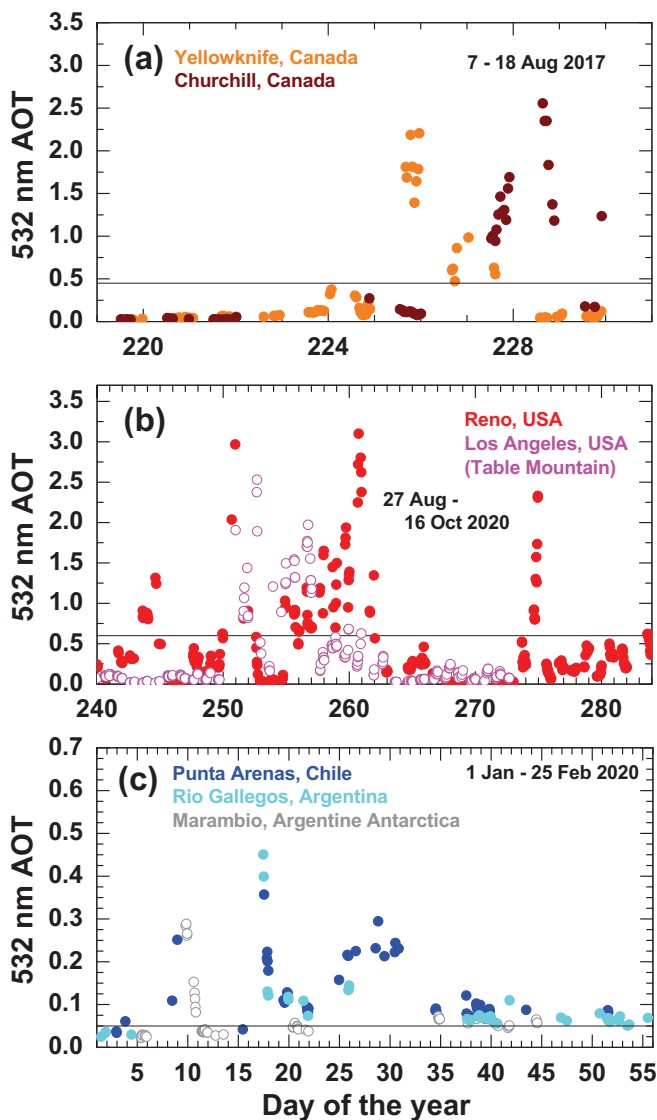

**Figure 3.** AERONET observations of strong smoke plumes in terms of 532 nm AOT: (a) optically dense stratospheric smoke layers over northern-central Canada after the major pyroCB-related fire event in British Columbia, Canada, in the afternoon of 12 August 2017 (day 224), (b) tropospheric smoke over the western United States during major forest fires in the late summer and early autumn of 2020, and (c) aged stratospheric smoke over southern South America and Arctica in January and February 2020 about 10000 km east of the Australian wildfires sources. The horizontal lines indicate the minimum AOT values considered in the determination of the conversion parameters. The smoke-free background 532 nm AOT levels are (a) 0.025-0.05, (b) 0.1-0.25, and (c) 0.025-0.035.

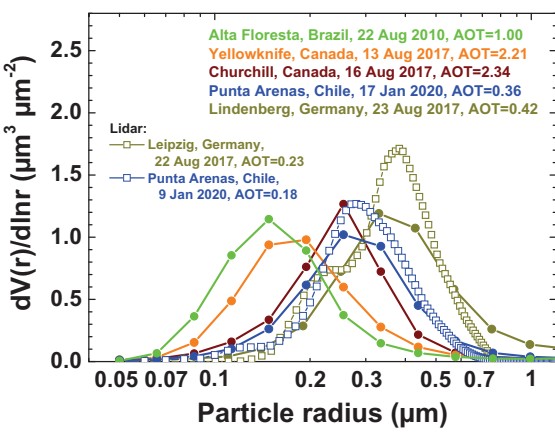

**Figure 4.** Comparison of normalized volume size distributions of smoke particles highlighting the shift of the size distribution towards larger particles with age of the observed smoke. The Amazonian smoke size distribution (Alta Floresta, green) is indicative for rather fresh smoke. Canadian smoke over Yellowknife (orange), Churchill (red), and Lindenberg (brown) was observed 1, 3-4, and 10-11 days after injection of smoke into the UTLS. The Punta Arenas observation (blue) was taken after about 8 days of long-range transport. The stratospheric size distributions obtained from lidar observations (open symbols, Punta Arenas, Leipzig) match well with the respective AERONET observations at Punta Arenas and Lindenberg (about 180 km northeast of Leipzig). The accumulation mode radius shifted from 150-200 nm (Yellowknife) to 300-400 nm (Lindenberg) within the 9-day travel of the 2017 smoke plumes from Yellowknife in Canada to Germany.

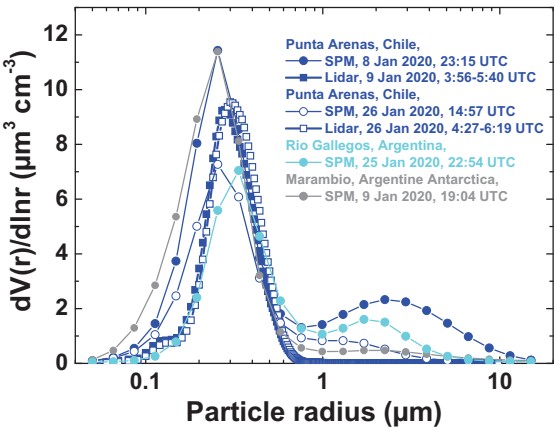

**Figure 5.** Normalized volume size distributions of smoke particles derived from column (tropospheric + stratospheric) AERONET sunphotometer (SPM) observations at Punta Arenas, Rio Gallegos, and Marambio in January 2020. In addition, size distributions obtained from the inversion of lidar-derived optical properties (squares) in the well-defined smoke layers are shown. Base and top heights of the smoke layers were 12.8 and 15.7 km on 9 January 2020 and 19.3 and 22.9 km on 26 January 2020, respectively. The lidar-derived size distributions show an accumulation mode only, a distinct coarse mode is absent.

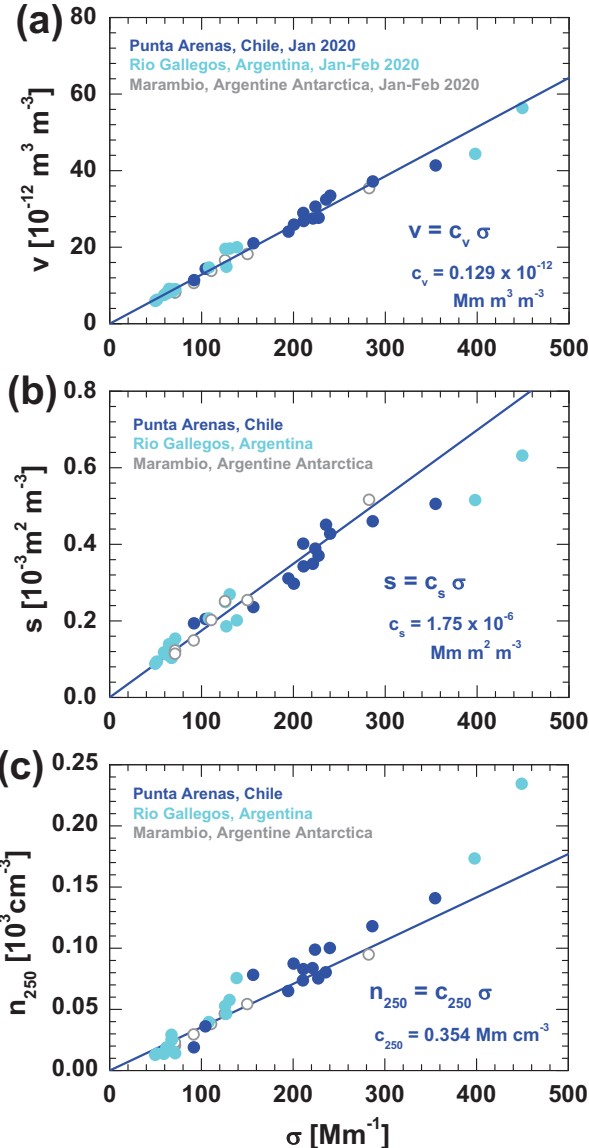

**Figure 6.** Relationship between smoke extinction coefficient $\sigma$ (532 nm) and (a) volume concentration $v$, (b) surface area concentration $s$, and (c) number concentration $n_{250}$ of aged stratospheric Australian smoke observed over the three AERONET stations in South America and Antarctica. The slopes are defined by the equations in the different panels a, b, anbd c. The conversion factors $c_v$, $c_s$, and $c_{250}$ in these equations are the mean values of the observed individual ratios of $v/\sigma$ (Eq. 19), $s/\sigma$ (Eq. 21), and $n_{250}/\sigma$ (Eq. 20). These mean values are given as numbers in the panels and together with the corresponding standard deviations also in Table 3.

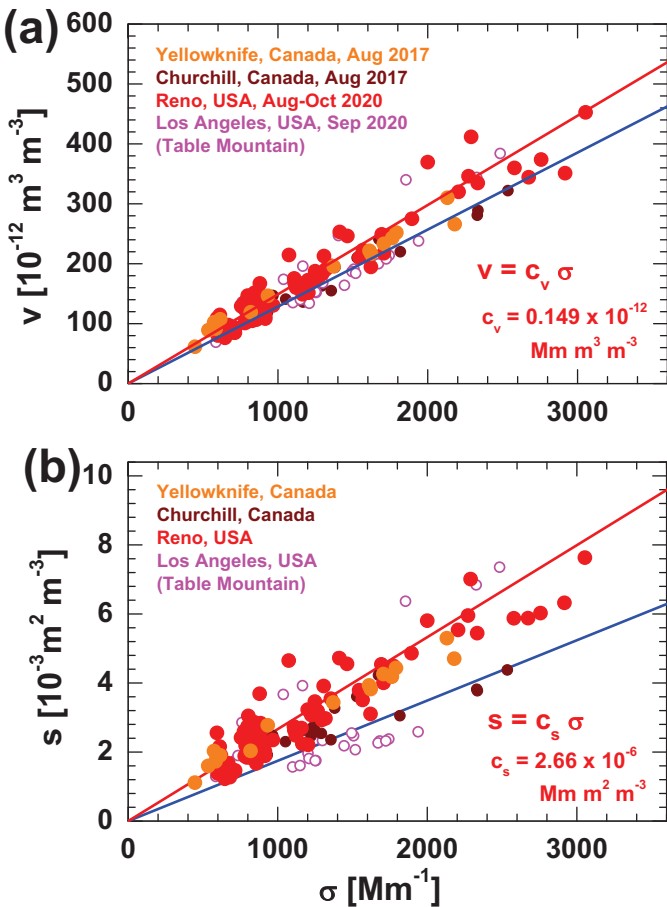

**Figure 7.** Same as Fig. 6a and b, except for fresh (Yellowknife) and aged stratospheric smoke (Churchill) in August 2017, and for mixtures of fresh and aged tropospheric smoke over Reno and Table Mountain, mostly observed in September and October 2020. The red lines are calculated with the equations given in panels a and b. They consider Yellowknife and Reno data, only. The conversion factors $c_v$ (Eq. 19) and $c_s$ (Eq. 21), again the mean values of all individual observations of the ratios $v/\sigma$ and $s/\sigma$, are given as numbers. The blue lines (taken from Fig. 6) are shown for comparison.

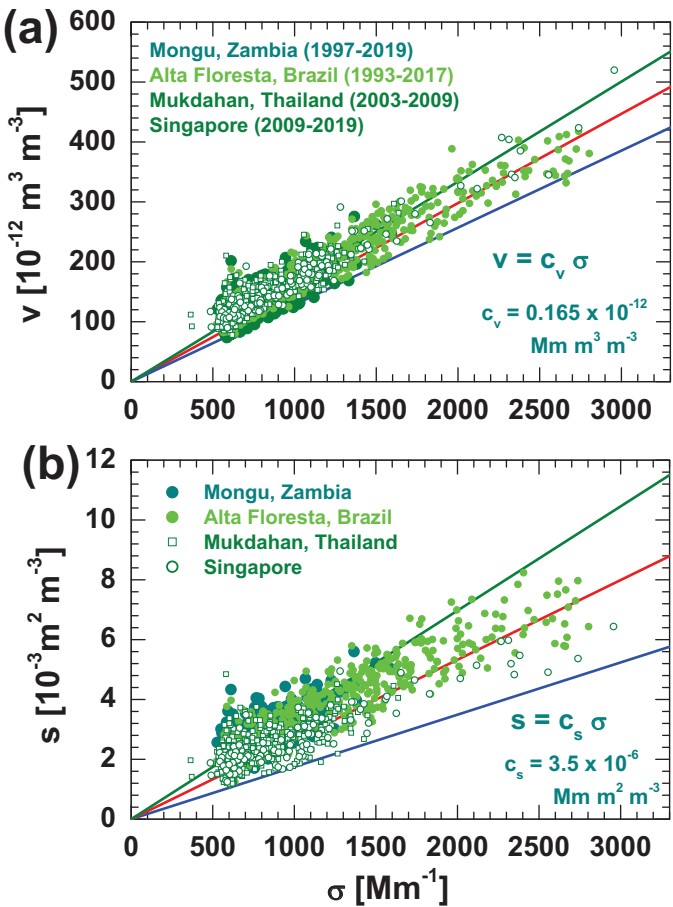

**Figure 8.** Same as Fig. 6a and b, except for African (Mongu), Amazonian (Alta Floresta), and Southeast Asian smoke (Mukdahan, Singapore) open olive circles for Mukdahan data, open olive squares for Sigapore data). The long-term, multiyear observations cover a wide range of burning material, fire conditions, and observations of fresh and aged smoke properties. The slopes (green lines, for the Mongu data set) are defined by the equations in the two panels a and b. The conversion factors $c_v$, and $c_s$ in these equations are the mean values of the observed individual ratios of $v/\sigma$ (Eq. 19) and $s/\sigma$ (Eq. 21). These mean values for the Mongu site are given as numbers. The blue and red lines (taken from Fig. 6 and 7) for aged Australian smoke (blue) and mixtures of fresh and aged North American forest fire smoke (red) are shown for comparison.

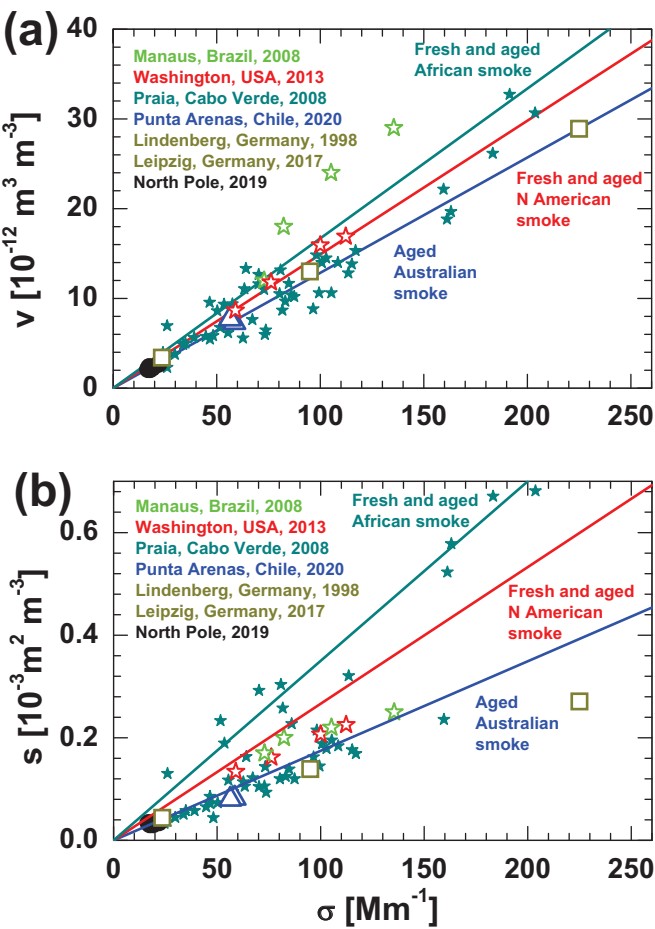

**Figure 9.** Same as Fig. 6a and b, except for a correlation between (a) lidar-derived $v$ and $\sigma$ and (b) lidar-derived $s$ and $\sigma$. The closed dark green stars indicate lidar observation of fresh and aged western African smoke taken in January and February 2008. The open green and red stars show lidar observations in Brazil and USA of mixtures of fresh and aged smoke during the summer seasons of 2008 and 2013, respectively. The two open blue triangles (Punta Arenas), three open squares (Lindenberg, Leipzig, Germany) and the black circles in the lower left corner (North Pole region) are representative for aged smoke. The thick blue, red and green AERONET-based lines show the mean increase of $v$ and $s$ with $\sigma$ for aged Australian smoke (blue), mixtures of fresh and aged North American forest fire smoke (red), and mixtures of fresh and aged southern African smoke (green).

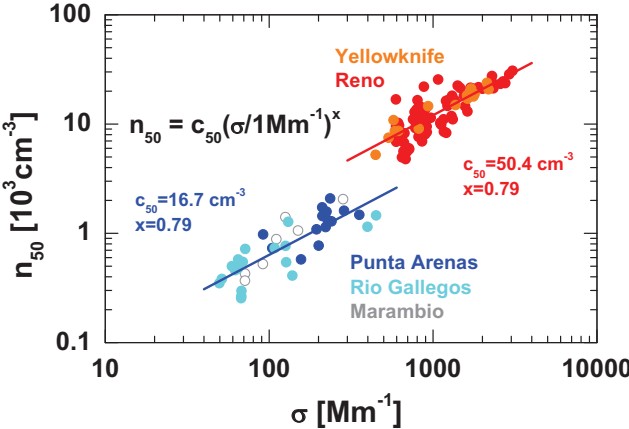

**Figure 10.** Relationship between smoke extinction coefficient $\sigma$ (532 nm) and particle number concentration $n_{50}$ for the combined Reno and Yellowknife data set (fresh and aged smoke) and the combined South American and Antarctic data set (aged smoke).

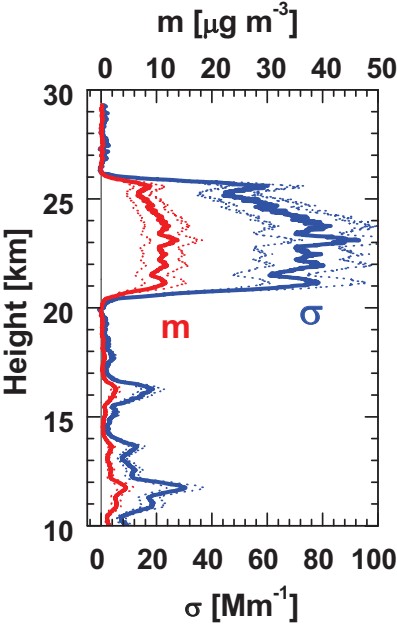

**Figure 11.** Smoke observation with lidar in the stratosphere over Punta Arenas on 29 January 2020 (see Fig. 1) in terms of the smoke extinction coefficient $\sigma$ and particle mass concentration $m$. Extinction coefficients were obtained by multiplying the respective backcatter coefficients with a lidar ratio of 95 sr. The errors margins (thin dotted) indicate relative uncertainties as given in Table 4 for the Raman lidar option in the case of aged smoke.

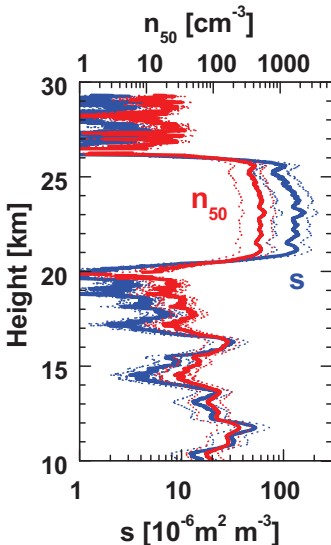

**Figure 12.** Retrieval results for 29 January 2020 in terms of surface area $s$ and particle number concentration $n_{50}$ (proxy for CCN) with error margins representing the uncertainties as given in Table 4 for the Raman lidar option in the case of aged smoke.

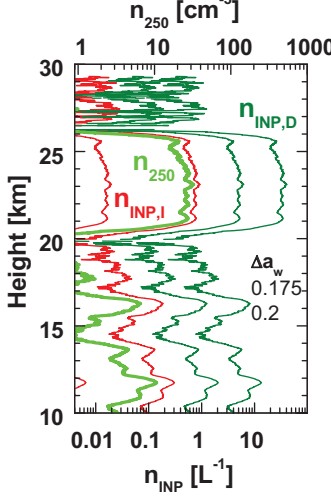

**Figure 13.** Retrieval results for 29 January 2020 in terms of INP concentrations $n_{\mathrm{INP,I}}$ and $n_{\mathrm{INP,D}}$ and large particle number concentration $n_{250}$ (considering particles with radius >250 nm). See text for more details of the INP computations in the case of immersion freezing (red profiles) and deposition nucleation (olive profiles). We consider Leonardite as the organic aerosol substance (see Table 2). The INP concentrations are estimated by assuming an air parcel lifting period of 600 s (period of supersaturation with $\Delta a_{\mathrm{w}}$ of 0.175 (low INP numbers) and 0.2 (high INP values)) and ice nucleation temperature of $-50°$C.

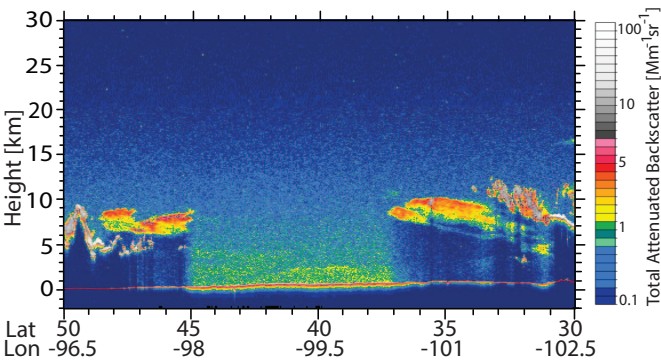

**Figure 14.** CALIPSO lidar observations of tropospheric smoke over North and South Dakota (45-48.5°N, 7-9 km height), and over Texas (33-37°N, 8-10 km height) on 13 September 2020, around 9:15 UTC (CALIPSO, 2020a). The smoke layers (in yellow to red) originated from British Columbia (North and South Dakota plume, travel time of 24 hours) and from California (Texas plume, 2-5 days of travel time) as HYSPLIT backward trajectories indicate (HYSPLIT, 2020).

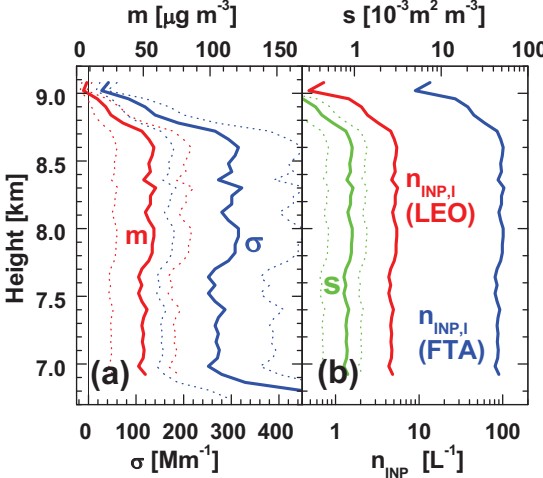

**Figure 15.** CALIPSO smoke observation in the stratosphere over North and South Dakota on 13 September 2020 in terms of (a) particle extinction coefficient $\sigma$ and mass concentration $m$, and (b) INP concentration estimates $n_{INP,I}$ for $T = -40°C$, $\Delta a_w = 0.2$, and two different organic substances (Leonardite, LEO, and free tropospheric smoke aerosol, FTA, see Table 2). The lidar-derived input parameter is the shown surface area concentration $s$. The CALIPSO aerosol backscatter coefficients were downloaded and averaged over the range from 45-48.5°N (CALIPSO, 2020b) and then multiplied with 70 sr to obtain the extinction coefficients. Error margins (thin dotted lines) show the uncertainties in $\sigma$, $m$, and $s$ as given in Table 4 (fifth column).