# Peer review of "Tropospheric and stratospheric wildfire smoke profiling with lidar: Mass, surface area, CCN and INP retrieval"

_Atmospheric Chemistry and Physics, 2020_

## Referee Comment (RC1) · Anonymous Referee #2 · 10 Dec 2020

The paper by Ansmann et al. describes a methodology to determine smoke particles' mass, volume and surface area concentrations and then the condensation nuclei (CCN) and ice-nucleating particles (INP). These features are determined for smoke layers in upper troposphere and lower stratosphere as measured by lidars. The estimates are based on profiles of particles backscatter coefficient retrieved from lidar measurements and a series of conversion coefficients determined from photometer (Aeronet) measurements. Two case studies are shown: for stratospheric smoke observations with Calipso and ground based lidar. In my opinion, the paper is acceptable for publication after minor revisions. Please address all the items discussed below.

[Figure]

General and specific comments

Pp3, l 2 (Introduction) and Section 3. Please make it clear that in the present study no polarization is used. Thus, it should be clearly stated that from Poliphon method you only use the formulas for conversion factors (based on Aeronet).

Pp7, l 19-21. If I understood correctly, the studies by Haarig and Ohneiser do not do any separation technique for the pollution layers (Poliphon like). So basically, it is just an assumption, eventually based on literature values for LR, depolarization. Thus, I guess that the statement "So we assume pure smoke layers when the smoke is detected in the stratosphere" is enough.

Pp 7, l 23-26: Why you did not use Poliphon for particles separation as you had de-polarization information (Fig 1 and Calipso)? Is it just because you assume dust is not present in UTLS? If so, Polihon method can be applied only in lower troposphere. Please clearly state it.

Pp 7, l 32 - pp 8, l 1-7: how do you chose 'an appropriate lidar ratio SL'? Basically, from Table 2 we have a range of 40-110 sr @ 532nm. Based on extensive literature review (39 papers reviewed) shown by Adam et al. (2020), LR at 532 ranged from 26 sr (aged smoke, Muller et al., 2005) to 147 sr (aged smoke, Mariano et al., 2010). Minimum and maximum values found for fresh smoke were 30 sr (Burton et al, 2012) and 100 sr (Mylonaki et al, 2017). Please see Table S3 in Adam et al (2020). Thus, it is not trivial how to choose an 'appropriate' value. Which are the criteria for your recommendations of LR range? Which uncertainty do you assume for them? Please comment more. I do not see using the colour ratio of the lidar ratio in this study and consequently, I am not sure if you should mention this (pp 8, l 5-7). From what I understand, this study is based on backscatter coefficient only. Thus, it is no need to mention LR and further CR of LR (they are not useful here).

References mentioned above:

Adam M. et al: Biomass burning events measured by lidars in EARLINET – Part 1: Data analysis methodology, 13905–13927, https://doi.org/10.5194/acp-20-13905-2020, https://doi.org/10.5194/acp-20-13905-2020-supplement, 2020.

Müller D. et al..: Raman lidar observations of aged Siberian and Canadian forest fire smoke in the free troposphere over Germany in 2003: Microphysical particle characterization, J. Geophys. Res., 110, D17201, https://doi.org/10.1029/2004JD005756, 2005.

Mariano G. L. et al.: Assessment of biomass burnings activity with the synergy of sunphotometric and LIDAR measurements in São Paulo, Brazil, Atmos. Res., 98, 486–499, https://doi.org/10.1016/j.atmosres.2010.08.025, 2010.

Burton S. P.: Aerosol classification using airborne High Spectral Resolution Lidar measurements – methodology and examples, Atmos. Meas. Tech., 5, 73–98, https://doi.org/10.5194/amt-5-73-2012, 2012.

Mylonaki M. et al: Aerosol optical properties variability during biomass burning events observed by the EOLE-AIAS depolarization lidars over Athens, Greece (2007-2016), 28th ILRC, Bucharest, Romania (2017).

Pp 8, l 15-16: how can we estimate the few percent BC in the smoke?

Pp 8, l 25: is it 500 nm or 50 nm?

Pp 9, l 8: you used n60 in Mamouri/Ansmann (2016). In that paper it is mentioned n50 in the abstract. In section 2.1 it is mentioned both. In 3.2 you determine n50=c60*sigma^x. In the current paper (table 1), you have n50=c50*sigma^x. Basically, c is different not n. Please rephrase. Maybe I did not understand (apologies for this).

Pp 12, l 20 (eq. 9): why did you use 440 and not 500? Why did you use 500 in Ansmann et al. (2019a)? Please argue.

Pp 13, l 7 (eq 12): as we use photometer data, it should be mean value of the Vj/tau_j (as in eq. 10). vj and sigma_j are characteristics for lidars only. It is similar for eqs

13 and 14. In my opinion it should be divided by tau_j while having N250 and S (as for column integrated). For eq. 16 I think it should be log(N50) versus log(tau_j). Shinozuka uses also AOD.

Pp 13, l 12: I do not see eq. 15.

Pp 14, l 21: please add date/time of the injection

Pp 14, l 30: how did you chose 0.45? why not 0.4 or 0.5? which criteria did you use?

Pp 15, l 2: same questions for 0.07

Pp 15, l 7: please add info about smoke layer geometry (from Table 1, Ohneiser) as well as about inversion method (i.e. did you use 3 backscatter and 2 extinctions?)

Pp 15, l 21: can you comment on the quantitative differences?

Pp 15, l 30: Figures. . .

Pp 16, l 1: I still don't understand the reason of dividing by D (here, 1000) unless there is a comparison with lidar retrievals. Please revise:

so that ïĄş (in Mm-1) divided by 1000 yield the basic AERONET 532 nm AOT value. Should be: so that ïĄş (in Mm-1) multiplied by 1000 yield the basic AERONET 532 nm AOT value.

Pp 16, l 17-18: you mention the larger spread because you plot both fresh and weakly aged. Why did you plot them together in the first place? Could you please plot them separately? Your statement does not hold for a) (both are spread). For b) and c) there is a better correlation for Yellowknife.

Pp 16, l 27. Why didn't you determine n50 for each fresh and weak aged in Fig. 9? Please add regression coefficients for both fresh and weak aged (besides their cumulative estimation).

Pp 16, l 1-5: how did you compute the conversion parameters (cv and cs) shown in

Figs 7 and 8 for fresh + weak aged? According to the numbers in Table 4, for the mixture f+wa, you have cv=0.181 (cs is the same in figures and table).

P 17, l5 (section 4.3): please clearly mention that the values in Table 4 correspond to the data shown in Figs 6-9. Thus, first row, for aged smoke, shows the result for data shown in Fig. 6 (and Fig 9 for n50). For fresh and weakly aged smoke, you have the results based on Fig. 7 data (and Fig 9 for n50). Please mention that f+wa is the average of each individual fresh and weak aged smoke. Please mention why you did not compute n50 for fresh only and for weak aged only. On table 4 and Fig 9 you show only for mixture.

Pp 17, l 10-11: Why is kext important to show? To show that it is higher for aged smoke? Ansmann et al 2012 does not contain kext for smoke so I am not sure it should be cited here. Several values of kext are given for different aerosol mixtures and different mean diameter of the particles in Horvath (chapter 16 of Atmospheric Particles, ed. Harrison and van Grieken, 1998, ISBN: 978-0-471-95935-9). For mixed absorbing particles, one can see the largest value for kext at 500 nm.

Pp 17, l 25: how is the "appropriate SL of 95 sr" determined? Is it from photometer AOD constraint? Earlier you mentioned (as for Table 2) a range of 65-80 sr at 532 nm. Please comment. Which error did you consider for SL when computing extinction error and further which error did you consider for density when computing M?

Pp 18, l 4: should be n250 instead of n50?

Pp 19, l 25: How did you select SL=80 sr?

Pp 19, l 26: you say you use conversion factors for fresh + weakly aged. In Fig 16 caption it is mentioned fresh only. Please correct. Why you did not show n50 for the two Calipso cases? According to Table 4, there is no n50 calculation for either fresh or weakly aged. There is a calculation only for fresh + weakly aged. As said before, why?

Pp 31, Table 1: change c100 with c50

Pp 33, Table 4: please add info for clarity: Yellowknife (for fresh smoke) and Churchill (for weakly aged smoke) (Figs 7 and 9) and from observations at Punta Arenas, CEILAP-RG, and Marambio (for aged smoke) (Figs 6 and 9). Latest row: I guess it is Lidar (Manaos). Use city instead of region (as for the others). As mentioned above, why you did not calculate c50 and x for fresh and weakly aged? Please calculate them and add them in the table.

Pp 38, Fig 6: change dust with smoke.

Pp 39: see comments above about cv. According to table 4 it should be cv=0.181.

Figs 7-8: I suggest deleting 'except' from 'except for' (which I understand it as "with the exception of"). I may be wrong so please check with a native English speaker. Why do we need fig 8? If you represent the lidar data on Fig 7 is to crowded? I guess it is an overlap of the lidar data (stars in Fig 8) with aged values in Fig. 7.

Figs. 6-9 and corresponding text: please make it clear that equations for cv, cs, c50, c250 (12-14, 16) are based on the data shown in Figs 6-9 (you mention Fig 2 – which has AOT). Thus, in Table 4 we have cv (and other coefficients) for aged smoke based on Fig 6 (and Fig 9 for n50). For cv and the others for fresh + weak aged we have 4 values based on Fig. 7-8 (and Fig 9 for n50).

Pp 43, Fig. 12: Sorry I missed it. What are the two profiles for each of INP representing? We see two red, blue and green profiles.

Pp 45, Fig. 16: used fresh conversion or fresh + weakly aged conversion?

General question: I did not revise deeply section 3 and thus I do not question all the parametrizations and various assumptions. I would like you to comment on the high uncertainty for n50, n250 and INP. We have uncertainties with a factor of 2-3 (i.e. 200 % - 300%). Please comment on CCN and INP values and their uncertainty as input in other studies (e.g. models). If I remember correctly, various models provide some variables with such high uncertainty.

---

## Referee Comment (RC2) · Anonymous Referee #1 · 19 Jan 2021

**Review of "Tropospheric and stratospheric wildfire smoke profiling with lidar: Mass, surface area, CCN and INP retrieval" by Ansmann et al., 2020**

This is one of a series of papers devoted to developing a robust set of conversion coefficients that can be used to extend the amount of information that can be derived from polarization-sensitive lidar measurements of specific aerosol types. A previous publication concentrated on dust and on separately assessing the contributions from dust and other types in aerosol mixtures. As indicated by the title, this latest work focuses on estimating microphysical properties of smoke.

This is fascinating paper that outlines several recent advances in a comprehensive and very ambitious retrieval scheme. The foundations of the retrieval algorithm are a collection of conversion coefficients derived from AERONET data and, to a lesser degree, from  $3\beta + 2\alpha$  lidar retrievals augmented by multi-wavelength depolarization ratio measurements. The authors' contention is that reliable estimates of aerosol microphysical properties such as mass concentration and surface area can be estimated by application of their conversion coefficients to single wavelength measurements made by polarization-sensitive elastic backscatter lidars. In principle, I believe they are correct. But in practice, the reliability of the estimated quantities will depend critically on the uncertainties in the underlying lidar measurements and in the conversion coefficients. The authors' technique has the potential to make a valuable contribution by extending the scope of information (and the subsequent science) that can be derived from existing and future lidar measurements, and hence this paper should eventually be published. But, prior to publication the authors should provide some more rigorous explanations and assessments of the uncertainties they ascribe to their conversion coefficients. In particular, (a) there should be some (perhaps minimal) discussion of random vs. systematic error with respect to the conversion coefficients, and (b) Table 1 should be revised so that it explicitly states the expected uncertainties (or range of uncertainties) associated with each measurement and/or conversion coefficient.

The remainder of this review consists of two parts. Immediately below, the authors will find several comments and questions about specific topics that particularly piqued my interest as I read their manuscript. Following this section I have attached an annotated version of the paper that includes a number of additional remarks that should be considered in a future revision.

**Specific Topics**

How broadly applicable are the authors' smoke conversion factors? Best I can tell, they have not characterized smoke from smoldering fires at all, but have instead focused on crown fires that inject unusually large masses of smoke into the stratosphere. I am surprised by the lack of attention given to the African smoke transported out over the southwest Atlantic Ocean, especially given the number of prior publications by many of the coauthors that focus on this yearly phenomenon.

I was similarly surprised to see no mention of multiple scattering effects in the lidar retrievals, and particularly in the analyses of the CALIOP data. (More on this later.)

In Section 3.1 (Part A: basic polarization lidar analysis), the different treatments of stratospheric and tropospheric smoke plumes seem problematic to me. Consider Figure 1 below, which shows CALIPSO volume depolarization ratio measurements for three distinct smoke plumes originating from the California wildfires in September 2020. All three plumes show elevated volume depolarization ratios. As shown by Gialitaki et al. (2020), spheroid and/or Chebyshev particle

shapes can explain the high depolarization ratios observed in the stratospheric smoke plumes generated by the 2017 Canadian wildfires. Why then should we attribute the high depolarization ratios in the tropospheric plumes in Figure 1 solely to soil dust rather than to some combination of dust and non-spherical "pure smoke" particles? More importantly, what are the consequences of this decision in the subsequent retrievals of microphysical properties? How much aging is required to create a sufficient number of non-spherical smoke particles to perturb the depolarization ratios? Please provide some additional discussion of this topic.

---

## Author Comment (AC1) · 1 Apr 2021

Dear Editor, Dear Reviewers!

First of all, many thanks to the reviewers for being interested, for careful reading, and for providing a lot of encouraging suggestions and recommendations. We accepted and considered almost all of the comments and became very much motivated to extend our efforts, as you will see below.

In this reply letter, answers to BOTH reviewers (#2 first, then #1) are included

Main points of changes are the following:

- We introduced a new structure (8 sections instead of 6 sections before). Sections 1-3 are not changed, the old Section 4 is now given in two parts (Sect. 4: AERONET sites and analysis, Sect. 5: AERONET results). And we introduced a new Sect.6 on 'Summary and Uncertainty'. But the general structure remained unchanged. The 8 sections give the full (long) paper a better easy-to-follow structure, we think.

- We increased the number AERONET stations from 5 to 11 to cover now also tropospheric smoke in North America (Reno, Table Mountain), the Amazon region (Alfa Floresta), southern Africa (Mongu, Zambia), and Southeast Asia (Mukdahan, Thailand, and Singapore). As recommended, we now show a map with all these 11 stations.

- Guided by the paper of Sayer et al., ACP 2014 (AERONET-based global smoke analysis to create a smoke model), we now concentrate entirely on the fine-mode aerosol products of the AERONET data base. We changed our software package accordingly. This approach considerably facilitated the entire smoke-related analysis, the results are more accurate, and biases, e.g., by PBL coarse aerosols is removed.

- Because of more stations, we introduce two new figures … with correlation results (and conversion parameters) for North Amercian smoke in Fig. 7, and for the Amazon region, southern African, and Southeast Asian smoke in Fig 8.

- We realized during the extended AERONET data analysis that we should just distinguish between two scenarios: (a) aged smoke (after long-range transport, far away from the smoke sources) and (b) mixtures of fresh + aged smoke (close to the smoke sources).
  In the majority of AERONET smoke observations close to the fire regions, fresh smoke is mixed with aged smoke. This background smoke accumulates over the fire regions during the long fire seasons. Thus, we skipped to derive a conversion scheme for fresh smoke.

- We introduced a new Section 6 with (a) an overview of all the conversion parameters (Table 3, based on AERONET data only now, no lidar-derived conversion parameters anymore) and (b) a detailed discussion on the uncertainties in all retrieval input parameters and in the smoke retrieval products (new Table 4).

- To compensate the increased length of the manuscript by introducing new stations and new results, we reduced the case studies (section 7). Besides the first case study (already given in the submitted version) we skipped the other two, but now consider a new tropospheric smoke case (exactly the case reviewer #2 suggested: CALIPSO overflight, 13 September 2020 from North and South Dakota down to Texas…). Now we have two really contrasting cases: Case 1: aged stratospheric smoke observed with ground-based Raman lidar, and Case 2: fresh tropospheric smoke observed with spaceborne lidar.

**Step by step reply to the comments.**

Our answers in blue, in the revised version, we show sections with significant changes in bold (only).

**Reviewer #2**

This is one of a series of papers devoted to developing a robust set of conversion coefficients that can be used to extend the amount of information that can be derived from polarization-sensitive lidar measurements of specific aerosol types. A previous publication concentrated on dust and on separately assessing the contributions from dust and other types in aerosol mixtures. As indicated by the title, this latest work focuses on estimating microphysical properties of smoke. This is fascinating paper that outlines several recent advances in a comprehensive and very ambitious retrieval scheme. The foundations of the retrieval algorithm are a collection of conversion coefficients derived from AERONET data and, to a lesser degree, from $3\beta + 2\alpha$ lidar retrievals augmented by multi-wavelength depolarization ratio measurements. The authors' contention is that reliable estimates of aerosol microphysical properties such as mass concentration and surface area can be estimated by application of their conversion coefficients to single wavelength measurements made by polarization-sensitive elastic backscatter lidars. In principle, I believe they are correct. But in practice, the reliability of the estimated quantities will depend critically on the uncertainties in the underlying lidar measurements and in the conversion coefficients. The authors' technique has the potential to make a valuable contribution by extending the scope of information (and the subsequent science) that can be derived from existing and future lidar measurements, and hence this paper should eventually be published.

*But, prior to publication the authors should provide some more rigorous explanations and assessments of the uncertainties they ascribe to their conversion coefficients. In particular, (a) there should be some (perhaps minimal) discussion of random vs. systematic error with respect to the conversion coefficients, and (b) Table 1 should be revised so that it explicitly states the expected uncertainties (or range of uncertainties) associated with each measurement and/or conversion coefficient.*

We removed Table 1 (submitted version). Instead, we introduced a new table (Table 4), exclusively filled with uncertainty estimates for all of the conversion input parameters and the smoke conversion products. To also include the uncertainties in the lidar observations (profiles of backscatter coefficients) , we distinguish three different lidar scenarios and different levels of uncertainties in the backscatter profiles: Raman lidar/HSRL, backscatter lidar at ground, backscatter lidar in space with a bit higher backscatter uncertainty compared to the ground-based lidar scenario. We introduced a new section (Section 6) to give these uncertainties and the related error discussion more room.

Specific Topics (part (a))

How broadly applicable are the authors' smoke conversion factors? Best I can tell, they have not characterized smoke from smoldering fires at all, but have instead focused on crown fires that inject unusually large masses of smoke into the stratosphere. I am surprised by the lack of attention given to the African smoke transported out over the southwest Atlantic Ocean, especially given the number of prior publications by many of the coauthors that focus on this yearly phenomenon.

The reviewer is right. So, motivated by this comment, we analyzed further AERONET data sets. We now include 6 additional AERONET stations to cover tropospheric smoke as well as subtropical and tropical regions: So, we added Reno, Table Mountain-CA, and the long-term data sets of Alta Floresta

(Amazon region), Mongu (Zambia), Mukdahan (Thailand), and Singapore in our analysis. These stations also cover different kinds of fires, i.e., flaming as well as smoldering fires, and many types of burning material. However, it is impossible to separate observations of smoke from flaming fires and from smoldering fires. It is even impossible to separate fresh-smoke observations (except for Yellowknife) from observations of mixtures of fresh and aged smoke. You always have some kind of mixtures of fresh and aged smoke. Only for aged smoke, the classification is quite simple and straight forward. When we observe smoke after intercontinental (long-range) transport it is definitely aged smoke. In the revised manuscript, we follow the analysis strategy of Sayer et al. (2014) who investigated AERONET smoke observations around the globe. Also they did not make any attempt to separate fresh from aged smoke, flaming fires from smoldering fires, or different biomass burning material.

I was similarly surprised to see no mention of multiple scattering effects in the lidar retrievals, and particularly in the analyses of the CALIOP data. (More on this later.)

There was no hint because we believed that multiple scattering effects can be ignored. Prata et al. (2017) stated that there should be a 5%-10% impact of multiple scattering at normal smoke conditions, e.g., when the smoke extinction coefficients are < 300 to 500 Mm-1. We mention the multiple scattering impact now in the (new) case study in section 7 and give reference to Prata et al. (2017).

In Section 3.1 (Part A: basic polarization lidar analysis), the different treatments of stratospheric and tropospheric smoke plumes seem problematic to me. Consider Figure 1 below, which shows CALIPSO volume depolarization ratio measurements for three distinct smoke plumes originating from the California wildfires in September 2020. All three plumes show elevated volume depolarization ratios. As shown by Gialitaki et al. (2020), spheroid and/or Chebyshev particle shapes can explain the high depolarization ratios observed in the stratospheric smoke plumes generated by the 2017 Canadian wildfires. Why then should we attribute the high depolarization ratios in the tropospheric plumes in Figure 1 solely to soil dust rather than to some combination of dust and non-spherical "pure smoke" particles? More importantly, what are the consequences of this decision in the subsequent retrievals of microphysical properties? How much aging is required to create a sufficient number of non-spherical smoke particles to perturb the depolarization ratios? Please provide some additional discussion of this topic.

We agree with all this and had a hard time to find an elegant way to avoid a long discussion on this here, in this paper, in which we just want to focus on the conversion of smoke optical into microphysical and cloud-relevant properties.

In the beginning of Sect. 3 we state:

Regarding the separation of smoke and dust fractions by means of the polarization lidar technique, we have to distinguish two branches. As long as the smoke-containing layers occur at low altitudes (in the lower and middle troposphere up to 5-7 km height), we can apply the traditional approach to determine the smoke fraction in dust-smoke mixtures by assuming a low smoke depolarization ratio of <0.05 and a high mineral dust depolarization ratio of 0.31. In the lower and middle troposphere, aging of the smoke particles is usually fast including the development of a spherical shape of the smoke particles. Furthermore most of the smoke particles are liquid (at least the shell) at comparably high temperatures and moisture levels. All this leads to a low smoke depolarization ratio at all laser wavelengths from 355 to 1064 nm.

However, if the smoke is lifted directly into the upper troposphere and lower stratosphere (UTLS), the smoke properties and aging features may be significantly different. With increasing height, and thus decreasing temperature, water vapor content, and amount of condensable gases, the aging

process is slow and the smoke particle become partly glassy. These effects seem to prohibit the development of a perfect spherical shape of the shells. As a consequence, the depolarization ratio can be as high as 0.15-0.2 at 532 nm at greater heights.However, we also observed low smoke depolarization ratios in the UTLS region as shown by Ohneiser et al. (2021}. Thus, in the case of UTLS smoke observations, the dust-smoke separation technique cannot be used. We have to assume that smoke layers are dominated by smoke (smoke fraction >0.9) in the UTLS regime.

As stated previously, I believe that perhaps the most valuable application of the POLIPHON algorithm will be in retrospectively analyzing the wealth of historical data acquired by groundbased, airborne, and space-based elastic backscatter lidars. However, that analysis cannot simply rely on assumed parameters derived by advanced lidars such as the Polly $3\beta + 2\alpha + 2\delta$ Raman systems, but must instead exploit the historical data to the fullest. The authors' analysis of the CALIOP data acquired 15 August 2017 over the Hudson Bay provides an illustrative example. Fairly early in their paper (Table 2), the authors take note of the large range of lidar ratios that can be found in smoke. For the Hudson Bay CALIOP data, they assume a lidar ratio of 80 sr. However, use of this value is not supported by the CALIOP measurements. As the authors note, this plume is frequently opaque at 532 nm between ~64.5°N and ~68.5°N. Furthermore, as seen in Figure 2, the CALIOP cloud-aerosol discrimination algorithm frequently misclassifies the smoke plume as cloud. However, irrespective of layer type, lidar ratios for opaque layers can be estimated using $S = 1 / (2 \times \eta \times \gamma')$ where $S$ is the lidar ratio, $\eta$ is the layer effective multiple scattering factor, and $\gamma'$ is the layer integrated attenuated backscatter. Young et al., 2018 (https://doi.org/10.5194/amt-11-5701-2018) describe an algorithm for optimizing the estimate of $S$ for a specified value of $\eta$.

Before we give a more complete answer below, just one question: How well do you know the multiple scattering impact in the case of opaque smoke layers? To our opinion, the impact is unkown.

Figure 3 shows optimized values of effective lidar ratio, $\eta S = 1 / 2\gamma'$ for the opaque portions of the smoke layer shown in Figure 2. The mean effective lidar ratio is 44.23 ± 3.57 sr. Assuming a multiple scattering factor of 1 (i.e., the value used in the CALIOP aerosol analyses), this lidar ratio would be on the low end (but not out of range) of the smoke lidar ratios reported in Table 2. Alternatively, one might posit a smoke multiple scattering factor of 44.23/80 ≈ 0.55, although this unlikely given that the value is ~10% lower than the mean multiple scattering factor (0.608 ± 0.004) assigned to the portions of this layer that are misclassified as clouds (see Garnier et al., 2015, https://doi.org/10.5194/amt-8-2759-2015, and Young et al., 2018). As seen in Figure 4, this difference in the multiple scattering factor makes a substantial difference in the magnitude of the particulate backscatter coefficients retrieved within the layer. While the effective lidar ratio is essentially constant through the along-track extent of the layer, the particulate backscatter coefficients within the profiles misclassified as cloud ($\eta \approx 0.608 \pm 0.004$) are, on average, ~25% higher than those in the profiles correctly classified as aerosol ($\eta = 1$). This uncertainty in the partitioning of the effective lidar ratio between multiple scattering factor and 'true' lidar ratio is alone larger than the total error ascribed to the backscatter coefficients in Table 1, and thus demonstrates the need to account for multiple scattering in the lidar retrievals. More generally, this example illustrates the need (expressed earlier) for a more rigorous accounting of the uncertainties ascribed to the other components of the POLIPHON algorithm.

The reviewer presented a nice case study. But as already mentioned, the multiple scattering impact is not well known in case of optically rather thick smoke layers. Effective or apparent lidar ratios around 45 sr may point to multiple scattering factors of 0.7 to 0.75 assuming that a smoke lidar ratio (single scattering) of 65 sr at 532 nm is most reasonable.

As a consequence of this discussion we decided to use another (new) CALIPSO case study and remove the two CALIPSO cases presented in the submitted manuscript. These two cases were

probably significantly affected by multiple scattering effects in an undefined way. We now use the 13 September 2020 observation (shown in the review of reviewer #2) in Section 7. On 13 September 2020, the CALIPSO lidar detected fresh tropospheric smoke over North and South Dakota and there were backscatter signals even from below the smoke layer, so CALIPSO observed a transparent smoke layer. The extinction values were about 300 Mm-1 for the two km thick smoke layer. In this case the multiple scattering impact should be low (multiple scattering factor of 0.95 is assumed).

Part (b): Comments of reviewer #2 in the PDF version of the manuscript:

We provide summarizing statements here only, not point-by-point answers.

Comments in the Abstract and Introductory:

We considered all the suggestions! One specific point should be addressed here: The problem of using column information (that includes PBL AOD and not only lofted smoke AOD) in smoke-related research is discussed later on in the paper. To avoid a significant impact of PBL aerosols (especially of coarse soil dust) we introduced threshold AOD levels of 0.45 and 0.6 (except for the very southern stations, Punta Arenas, Rio Gallegos, Marambio). Furthermore, we now consider the fine mode AERONET products only so that the impact of PBL coarse mode AOD contributions is no longer a problem. In cases of the southernmost stations, we checked all particle size distribution and left only observations with well pronounced smoke accumulation mode in the data set for the further use. Finally, we compared our findings with the ones of Sayer et al. (2014) (they have tables with values for the ratio of smoke column volume concentration to smoke AOD, for the fine mode spectrum) and found a generally good agreement with the results of Sayer et al (2014). So, we are convinced that our derived conversion parameter sets cover well the smoke characteristics observable around the world.

Comments to Section 3: Method and uncertainties:

We avoid to mention POLIPHON (throughout the manuscript). We fully concentrate on the conversion of optical into microphysical properties only. However, we briefly discuss that methods are available to identify and basically classify smoke layers and provide related references.

We now give an extended discussion on the uncertainties in the lidar ratio input. And more general, we have an extended uncertainty discussion in Section 6. We present a new Table 4 with all the uncertainty information, for all retrieval input parameters and the main retrieval products.

The reviewer wants more details to the status of the applicability of smoke INPs. We mention that we just present first DEMONSTRATION CASES of applications. The reviewer asked us to add more details what is needed to go on towards well characterized data products with reliable uncertain estimates and include some remarks of the expected complexity.

We now state at the end of Section 3.1: ... it remains to be emphasize that we put together several INP parameterizations in Sect. 3.1on for demonstration purposes. The research on the smoke impact on atmospheric ice formation is ongoing (Knopf et al., 2018). Presently, uncertainties in the prediction of $J_{\rm het,I}$ and $J_{\rm het,D}$ for organic aerosols are very high (Wang and Knopf, 2011, China et al., 2017}. However, the procedures introduced above allow us to estimate INP concentration profiles for organic aerosols and to study the potential impact of wildfire smoke on ice formation in tropospheric mixed-phase and ice clouds. In the upcoming years, strong field activities are required including comparisons of airborne in situ with lidar observations of smoke INP concentrations as successfully performed in the case of Saharan dust (Schrod et al., 2017, Marinou et al., 2019) and so-called cirrus closure experiments as realized in the case of cirrus formation in pronounced Saharan dust layers (Ansmann et al., 2019b) in order to check the applicability of

developed smoke INP parameterizations and to quantify the uncertainties in the INP estimates under real-world meteorological, cloud, and aerosol conditions. A first closure study with respect to smoke-cirrus interaction was recently presented by Engelmann et al. (2020).

Comments to Section 4: AERONET data, sites, and analysis procedure:

How is smoke separated in the case of complex mixture…..

We avoid to include complex mixtures (and thus strong interference by dust and haze). First of all, we carefully selected AERONET stations located in areas with strong fire activity. Then, we use observations with high AODs during the main fire months only. In the case of the clean southern stations (Punta Arenas, Rio Gallegos, Marambio) we carefully checked all size distributions and accepted only the observation with clear smoke impact.

We now provide clear information regarding used data and AERONET stations. We included a map showing all stations, as recommended. We clearly state how we selected the smoke observations (by using AOD threshold values and checking the size distributions), and, as a new point, we analyzed the fine mode products only. All this helps to have, at the end, a well selected set of smoke conversion factors.

We cannot separate flaming and smoldering fire types. Impossible! We even cannot separate fresh and aged in most of the observations. Only in cases with intercontinental aerosol transport we can be sure: This is aged smoke. In all other cases, we have a mixture of fresh and aged smoke (except for Yellowknife). This is now discussed several times.

**Reviewer #1**

The paper by Ansmann et al. describes a methodology to determine smoke particles' mass, volume and surface area concentrations and then the condensation nuclei (CCN) and ice-nucleating particles (INP). These features are determined for smoke layers in upper troposphere and lower stratosphere as measured by lidars. The estimates are based on profiles of particles backscatter coefficient retrieved from lidar measurements and a series of conversion coefficients determined from photometer (Aeronet) measurements. Two case studies are shown: for stratospheric smoke observations with Calipso and ground based lidar. In my opinion, the paper is acceptable for publication after minor revisions. Please address all the items discussed below.

General and specific comments

Pp3, l 2 (Introduction) and Section 3. Please make it clear that in the present study no polarization is used. Thus, it should be clearly stated that from Poliphon method you only use the formulas for conversion factors (based on Aeronet).

We follow this suggestion. We briefly discuss where the polarization lidar method can be used (in the lower to middle troposphere and where it cannot be used (upper troposphere and lower stratosphere, UTLS, see Section 3, page7). The word 'POLIPHON' is removed at all.

Pp7, l 19-21. If I understood correctly, the studies by Haarig and Ohneiser do not do any separation technique for the pollution layers (Poliphon like). So basically, it is just an assumption, eventually based on literature values for LR, depolarization. Thus, I guess that the statement "So we assume pure smoke layers when the smoke is detected in the stratosphere" is enough.

We rephrased many text parts, but the basic idea remained: We assume pure smoke when we detect smoke-containing layers at great heights (upper troposphere and lower stratosphere, UTLS)). Again, as we discussed above (reply to reviewer #2) and also in Section 3 (in the beginning), we cannot use

the separation technique when we cannot be sure that smoke depolarization ratio is <0.05, and we know meanwhile that smoke can produce depolarization ratios up to 0.2 at 532nm in the UTLS range.

Pp 7, l 23-26: Why you did not use Poliphon for particles separation as you had depolarization information (Fig 1 and Calipso)? Is it just because you assume dust is not present in UTLS? If so, Poliphon method can be applied only in lower troposphere. Please clearly state it.

As just mentioned, we cannot be sure that smoke depolarization ratio is <0.05 (as assumed in the POLIPHON method) in the UTLS height range. Because of typically fast lifting the smoke particles keep their original non-spherical shape in the dry ULTS environment for a long time. This is the reason that we cannot make use of the POLPIPHON method at greater heights and need to assume that the non-smoke contributions (e.g., by soil dust) to the smoke layers in the UTLS region is negligible. We discuss these aspects now in the beginning of Section 3. The main goal of the paper is to introduce the smoke conversion procedure. All this is now more clearly stated.

Pp 7, l 32 - pp 8, l 1-7: how do you chose 'an appropriate lidar ratio SL'? Basically, from Table 2 we have a range of 40-110 sr @ 532nm. Based on extensive literature review (39 papers reviewed) shown by Adam et al. (2020), LR at 532 ranged from 26 sr (aged smoke, Muller et al., 2005) to 147 sr (aged smoke, Mariano et al., 2010). Minimum and maximum values found for fresh smoke were 30 sr (Burton et al, 2012) and 100 sr (Mylonaki et al, 2017). Please see Table S3 in Adam et al (2020). Thus, it is not trivial how to choose an 'appropriate' value. Which are the criteria for your recommendations of LR range? Which uncertainty do you assume for them? Please comment more. I do not see using the colour ratio of the lidar ratio in this study and consequently, I am not sure if you should mention this (pp 8, l 5-7). From what I understand, this study is based on backscatter coefficient only. Thus, it is no need to mention LR and further CR of LR (they are not useful here).

First of all, we checked the Adam et al. (2020) paper very carefully, and guided by this paper we started to update the discussion on smoke lidar ratios (see Section 3, after the paragraph with the four equations). We updated Table 1 (now include the LR value range given by Mylonaki et al, 2017, and also include the LR range given by Tesche et al., 2011 for western Africa smoke).

At the end of the updated discussion, it is now clear what smoke lidar ratio should be used and how large the remaining uncertainty by this assumption in the retrieval is. All this is discussed again in Section 6 (summary and uncertainties), and summarized in Table 4, will all uncertainties in the retrieval input parameters and products.

Pp 8, l 15-16: how can we estimate the few percent BC in the smoke?

We cannot estimate the BC fraction. We changed the text. The message is just that we can have BC fractions from 2-10%, or even more, say 15%. But these values do not influence the smoke particle density estimate much.

Pp 8, l 25: is it 500 nm or 50 nm?

Diameter… 500 nm

Pp 9, l 8: you used n60 in Mamouri/Ansmann (2016). In that paper it is mentioned n50 in the abstract. In section 2.1 it is mentioned both. In 3.2 you determine n50=c60*sigmaˆx. In the current paper (table 1), you have n50=c50*sigmaˆx. Basically, c is different not n. Please rephrase. Maybe I did not understand (apologies for this).

We hope we totally removed this confusion in the revised version. Now, we just discuss n50, c50, and extinction exponent x. Nothing else. This is justified because smoke layers are typically dry (relative

humidity is <60%) so that aerosol water uptake effects are negligible. High humidity and water uptake needs to be considered in the case of moist PBLs, then we need n60 etc. ….

Pp 12, l 20 (eq. 9): why did you use 440 and not 500? Why did you use 500 in Ansmann et al. (2019a)? Please argue.

In Section 4.2 (on page 14), we now clearly state that the AERONET files we downloaded contain AODs for wavelength of 440, 675, 870, and 1020 nm, only. These AODs are computed from the derived  size distribution data sets (and the refractive index properties). So these AODs belong most directly to the size distributions, we use. Years ago, we downloaded also the observed (basic) AODs (including 500 nm AODs). We found that the difference between the two ways to obtain 532 nm AOD (starting from 440 nm or 500 nm) is very small, and has no impact of the conversion factor determination. So, we switched to 440nm to keep the data analysis simple.

Pp 13, l 7 (eq 12): as we use photometer data, it should be mean value of the Vj/tau_j (as in eq. 10). vj and sigma_j are characteristics for lidars only. It is similar for eqs 13 and 14. In my opinion it should be divided by tau_j while having N250 and S (as for column integrated). For eq. 16 I think it should be log(N50) versus log(tau_j). Shinozuka uses also AOD.

Yes, you are right! But we introduced v, s, n50, n250  and sigma to switch to concentrations and extinction vales just a few sentences before, and do not want to step back again. Since we divided all AERONET values by the arbitrary layer depth D, this is ok. If we would step back to column values and AOds, many readers would be probably confused.

Pp 13, l 12: I do not see eq. 15.

We removed this line.

Pp 14, l 21: please add date/time of the injection.

That is impossible! There were so many pyroCB events from 31 Dec 2019 to 2 January 2020 that we only made the attempt to find out (via HYSPLIT trajectories) how long the transport from southeastern Australia to Punta Arenas lasted. In this way, we got the numbers on travel duration: 8-12 days. And this information is sufficient for our study. We know we observed well-aged smoke.

Pp 14, l 30: how did you chose 0.45? why not 0.4 or 0.5? which criteria did you use?

In this specific case, we wanted to include all data for Yellowknife after the arrival of the  huge smoke plume at this station, as shown in Figure 3a (in the revised manuscript), and therefore we had to set 0.45.

Pp 15, l 2: same questions for 0.07

The background AOD is 0.025 to 0.035 at these clean stations (with probably 0.01 AOD contribution by coarse marine particles). By using the threshold of 0.07 (and checking all size distributions for a dominant smoke accumulation mode), we could consider a sufficient number of smoke observations in the correlation. In the revised manuscript, we could even go down with the threshold to 0.05 because now we consider the fine-mode AERONET products only (so that the marine coarse mode part is no longer a problem).

Pp 15, l 7: please add info about smoke layer geometry (from Table 1, Ohneiser) as well as about inversion method (i.e. did you use 3 backscatter and 2 extinctions?)

We added the information (use 3 backscatter and 2 extinction values in the lidar inversion method, except in the case of the Lindenberg 1998 field campaign, six backscatter wavelengths, Wandinger et al., 2002) in the text (at the end of Section 4.1) and provide the information of the layer top and base height in the caption of Fig. 5.

Pp 15, l 21: can you comment on the quantitative differences?

'Quantitative' was probably triggered because we used 'qualitatively'. Now we just state: the size distributions agree well. It is not the topic of the paper to analyze the agreement in quantitative details.

Pp 15, l 30: Figures: : :

We leave it open to the ACP language editors. I would also prefer: Figures…. But they often prefer Figure 6 and 7 and 8…

Pp 16, l 1: I still don't understand the reason of dividing by D (here, 1000) unless there is a comparison with lidar retrievals. Please revise: so that ïA¸s¸ (in Mm-1) divided by 1000 yield the basic AERONET 532 nm AOT value. Should be: so that ïA¸s¸ (in Mm-1) multiplied by 1000 yield the basic AERONET 532 nm AOT value.

To avoid too much confusion, we removed all these unnecessary explanations. We just introduce D to switch from column values to concentrations and from AODs to extinction coefficients.

Pp 16, l 17-18: you mention the larger spread because you plot both fresh and weakly aged. Why did you plot them together in the first place? Could you please plot them separately? Your statement does not hold for a) (both are spread). For b) and c) there is a better correlation for Yellowknife.

We changed our entire 'philosophy'. In that way, we simplified the full approach. After introducing the new stations (two additional stations in North American, and four stations in the Amazon region, southern Africa, and Southeast Asia, partly with  decades of smoke observations) we decided to show ALL data of these new stations (and of Yellowknife and Churchill) at the same time (no separation in fresh, weakly aged, or aged anymore). The cloud of scattered data (especially for the new stations with 10-20 years of data) indicates cases with dominating fresh smoke (upper part of the data field) and aged smoke cases (lower part of the data field), and that most of the cases are just mixtures of fresh and aged smoke. The unchanged part of the revised manuscript is the analysis of the southernmost observations at Punta Arenas, Rio Gallegos, and Marambio. Here, we had the unique chance to definitely develop the parameterization for aged smoke which we can now use for all cases with intercontinental smoke transport.

Pp 16, l 27. Why didn't you determine n50 for each fresh and weak aged in Fig. 9? Please add regression coefficients for both fresh and weak aged (besides their cumulative estimation).

We would like to leave it as it is. This is just an example how we obtain c50 and x. We are happy to visualize that we can have  quite different results for more fresh and for well-aged smoke. This is the basic message of the figure, not more. Note also, that the numbers are even higher for the southern African and Southeast Asian stations (Table 3). We provide some numbers of n50 (for the extinction coefficient of 100 Mm-1) in Section 5.6 for the different smoke observations, to highlight the contrast,  635 cm-3 (Punta Arenas,…), 1900 cm-3 (Yellowknife,…), 3200 cm-3 (Mongu, Zambia).

Pp 16, l 1-5: how did you compute the conversion parameters (cv and cs) shown in Figs 7 and 8 for fresh + weak aged? According to the numbers in Table 4, for the mixture f+wa, you have cv=0.181 (cs is the same in figures and table).

All this is now changed. In the revised version, we clearly state, how we computed all the different conversion parameters, now in Table 3 (by using Eq. (XX) and the data in Fig. YY).

P 17, l5 (section 4.3): please clearly mention that the values in Table 4 correspond to the data shown in Figs 6-9. Thus, first row, for aged smoke, shows the result for data shown in Fig. 6 (and Fig 9 for n50). For fresh and weakly aged smoke, you have the results based on Fig. 7 data (and Fig 9 for n50). Please mention that f+wa is the average of each individual fresh and weak aged smoke. Please mention why you did not compute n50 for fresh only and for weak aged only. On table 4 and Fig 9 you show only for mixture.

Again, we tried to improve the description how we got all the final numbers in Table 3 (in the revised manuscript).

Pp 17, l 10-11: Why is kext important to show? To show that it is higher for aged smoke? Ansmann et al 2012 does not contain kext for smoke so I am not sure it should be cited here. Several values of kext are given for different aerosol mixtures and different mean diameter of the particles in Horvath (chapter 16 of Atmospheric Particles, ed. Harrison and van Grieken, 1998, ISBN: 978-0-471-95935-9). For mixed absorbing particles, one can see the largest value for kext at 500 nm.

Since we do not need it here, we removed kext!

Pp 17, l 25: how is the "appropriate SL of 95 sr" determined? Is it from photometer AOD constraint? Earlier you mentioned (as for Table 2) a range of 65-80 sr at 532 nm. Please comment. Which error did you consider for SL when computing extinction error and further which error did you consider for density when computing M?

In this specific case, we were able to use our own Raman lidar observation of the lidar ratio at Punta Arenas. We state that now in Section 7.1. Regarding the uncertainties in all input parameters, we introduced a new section (Section 6) and the new Table 4. For the particle density we used 1.15 g m-3 and the relative error was assumed to be 20%. For the lidar ratio LR, we assume an uncertainty of 20% when we have a Raman lidar or HSRL so that we can measure LR, and an uncertainty of 35% when we have to estimate LR and use 70 sr at 532 nm.

Pp 18, l 4: should be n250 instead of n50?

n50 is ok! We extended the discussion on the importance of n50 estimates. We included aspects of a potential smoke impact on PSC formation and ozone depletion (see Section 7.1).

Pp 19, l 25: How did you select SL=80 sr?

Now we use a smoke lidar ratio of 70 sr in the new case study (Section 7.2, CALIPSO observation, 13 September 2020). In Section 3, we recommended to use 70 sr (as the best compromise, if nothing is known). The relative uncertainty in this estimate is set to 35% in the error analysis (see Table 4).

Pp 19, l 26: you say you use conversion factors for fresh + weakly aged. In Fig 16 caption it is mentioned fresh only. Please correct. Why you did not show n50 for the two Calipso cases? According to Table 4, there is no n50 calculation for either fresh or weakly aged. There is a calculation only for fresh + weakly aged. As said before, why?

As mentioned, we changed our 'philosophy'…. One result of these changes are the recommended conversion parameter data sets (in the lower part of Table 3). We distinguish smoke observations far away from the smoke source region (i.e., for aged smoke) and for smoke observations, close to the smoke sources (i.e., for mixtures of fresh and aged smoke). We studied the trajectories for case 2

(CALIPSO observations) and they pointed to fires in western Canada. According to the trajectories the smoke was one day old. But who knows? The fresh smoke was probably embedded in all the aged smoke from the surroundings (a large area in western North America was burning in September 2020). So, to our opinion, from the practical point of view it makes no sense to create a conversion parameter data set for very fresh smoke. More useful are conversion parameters for observations near smoke source regions. This is especially useful for CALIPSO overflights over the Amazon region or the huge fire areas in southern Africa or Southeast Asia.

Pp 31, Table 1: change c100 with c50

We changed that but at the end we removed Table 1 (of the submitted version).

Pp 33, Table 4: please add info for clarity: Yellowknife (for fresh smoke) and Churchill (for weakly aged smoke) (Figs 7 and 9) and from observations at Punta Arenas, CEILAP-RG, and Marambio (for aged smoke) (Figs 6 and 9). Latest row: I guess it is Lidar (Manaos). Use city instead of region (as for the others). As mentioned above, why you did not calculate c50 and x for fresh and weakly aged? Please calculate them and add them in the table.

We changed and re-organized the result part significantly after introducing the 6 new AERONET stations. We compare the lidar observations with the AERONET observations now separately in the new subsection 5.5. We removed all lidar-derived conversion parameters from Table 3 (in the revised version, was Table 4 in the submitted version). This mixing was confusing. And we do not make use of the lidar-derived conversion parameters.

Pp 38, Fig 6: change dust with smoke.

Done

Pp 39: see comments above about cv. According to table 4 it should be cv=0.181.

The updated Table 3 (Table 4 in the submitted version) should now be in full consistency with the related figures.

Figs 7-8: I suggest deleting 'except' from 'except for' (which I understand it as "with the exception of"). I may be wrong so please check with a native English speaker.

We leave that open to the ACP language editors. They have their own ideas.

Why do we need fig 8? If you represent the lidar data on Fig 7 is to crowded? I guess it is an overlap of the lidar data (stars in Fig 8) with aged values in Fig. 7.

We improved this part of the manuscript. Now we have Figure 6 (for the southernmost stations), Figure 7 (for the North American stations), and Figure 8 (for the Amazonian, African, Asian stations). Here we present AERONET data ONLY  (no lidar data anymore). And then we introduced a new subsection 5.5 with Figure 9 with all the lidar data. Although quite noisy, the lidar-based correlations between microphysical and optical smoke properties provide an independent way to obtain conversion parameters, and thus allow us to check the general quality of our AERONET based approach. And the lidar vs AERONET comparisons is fine and supports our approach.

Figs. 6-9 and corresponding text: please make it clear that equations for cv, cs, c50, c250 (12-14, 16) are based on the data shown in Figs 6-9 (you mention Fig 2 – which has AOT). Thus, in Table 4 we have cv (and other coefficients) for aged smoke based on Fig 6 (and Fig 9 for n50). For cv and the others for fresh + weak aged we have 4 values based on Fig. 7-8 (and Fig 9 for n50).

We follow this suggestion. We state clearly what data we used and how we calculated the mean conversion factors (Sections 5.2-5.4).

Pp 43, Fig. 12: Sorry I missed it. What are the two profiles for each of INP representing? We see two red, blue and green profiles.

The old Figure 12 (in the submitted version) is now Figure 13. To simplify the figure, we removed two curves (for the free tropospheric smoke aerosol type, FTA). Now we show only the immersion-freezing INP and the deposition-nucleation INP profiles (for just one organic aerosol substance, Leonardite), and this for two different humidity (or supersaturation) conditions. So, we have four curves, and another one showing n250.

Pp 45, Fig. 16: used fresh conversion or fresh + weakly aged conversion?

Figure 16 (submitted version) is replaced by a new one (Figure 15, showing a case with comparably fresh tropospheric smoke from western Canada). Now we clearly mention what conversion parameter set we used (we used the recommended conversion parameters for observations close to the fire regions, i.e., for the mixture of fresh and aged smoke).

General question: I did not revise deeply section 3 and thus I do not question all the parametrizations and various assumptions. I would like you to comment on the high uncertainty for n50, n250 and INP. We have uncertainties with a factor of 2-3 (i.e. 200% - 300%). Please comment on CCN and INP values and their uncertainty as input in other studies (e.g. models). If I remember correctly, various models provide some variables with such high uncertainty.

We discuss all uncertainties, including the ones for CCN and INP concentrations, in the new Section 6. But we did not include CCN and INP uncertainties in the new Table 4.

We state in Section 6: Uncertainties in the estimates of CCN and INP concentrations are not listed in Table 4. Comparison with airborne in situ observations of CCN profiles suggest that the uncertainty in the lidar-based CCN estimation is around 50%, and in extreme cases up to a factor of 2 (Duesing et al., 2018, Haarig et al., 2019, Genz et al., 2020). In the case of INP estimations, it is too early for an in-depth uncertainty analysis. A considerable number of dedicated field campaigns and further laboratory studies are needed before a trustworthy quantification of uncertainties in the INP estimation.

And at the end of Section 3.1. we wrote: …  it remains to be emphasize that we put together the INP parameterizations in Sect. 3.1 for demonstration purposes. The research on the smoke impact on atmospheric ice formation is ongoing (Knopf et al., 2018).. Presently, uncertainties in the prediction of $J_{\rm het,I}$ and $J_{\rm het,D}$ for organic aerosols are very high (Wang and Knopf, 2011, China et al., 2017}. However, the procedures introduced above allow us to estimate INP concentration profiles for organic aerosols and to study the potential impact of wildfire smoke on ice formation in tropospheric mixed-phase and ice clouds. In the upcoming years, strong field activities are required to investigate the role of wildfire smoke particles in ice formation processes in the UTLS regime. More field  studies  as conducted by China et al. (2017), comparisons of airborne in situ and lidar observations of smoke INP concentrations as successfully performed in the case of Saharan dust (Schrod et al., 2017, Marinou et al., 2019) as well as so-called cirrus closure experiments as realized in the case of cirrus formation in pronounced Saharan dust layers (Ansmann et al., 2019)  are required to improve our knowledge about the  influence of smoke on ice cloud evolution under real-world meteorological and aerosol conditions and to develop trustworthy and accurate smoke INP parameterizations. A first closure study with respect to smoke-cirrus interaction was recently presented by Engelmann et al. (2020).

---

## Author Comment (AC2) · 1 Apr 2021

In the attached document, all answers to the comments of the two reviewers are given. We provide an overview of all important changes first. Then the step-by-step answers are given, starting with the reply to review #2, followed by the reply to review #1

Please also note the supplement to this comment:
https://acp.copernicus.org/preprints/acp-2020-1093/acp-2020-1093-AC2-supplement.pdf

---

## Referee Report (RR1)

Review of "Tropospheric and stratospheric wildfire smoke profiling with lidar: Mass, surface area, CCN and INP retrieval", manuscript version 3, by Ansmann et al., 2021

This latest draft represents a prodigious rewrite of the authors' original manuscript. In it they address all of the major criticisms I noted in my initial review. By reorganizing the structure of the paper, they have also expanded their exposition in several places where I thought additional details would be particularly helpful. In my opinion, the revised manuscript can and should be published. Nevertheless, below I offer a few minor comments, questions, and suggestions. While addressing these is entirely optional, I believe that doing so might help further clarify a few of the points made in the paper.

Page 8, equations 1 through 4 : the symbol $\beta$ has not yet been associated with backscatter. Ideally, both $\beta$ and $L$ would be defined in the paragraph immediately following the first use of the symbols.

Page 8 : the authors recommend default values for smoke lidar ratios "if there is no possibility to obtain actual lidar ratio information from Raman lidar or High Spectral Resolution Lidar (HSRL) observations". Do they consider the lidar ratios obtained via constrained solutions of elastic backscatter lidar measurements (e.g., Prata et al., 2017) to be of insufficient accuracy? A comment on this would be helpful for potential users of the authors' technique.

Page 9, line 23 through page 12, line 7 : I did not revisit this material, as I assumed it was essentially unchanged from the first version of the paper.

Page 12, lines 8–19 : I expect readers attempting to implement the authors' technique will appreciate the addition of these caveats.

Pages 12–14, Section 4.1 : I'm especially pleased to see the inclusion of the AERONET data from the sites in Alta Floresta, Mongu, Mukdahan, and Singapore.

Page 16, line 23 : Since the authors are presenting an "AERONET-based correlation analysis", I'm more than a little surprised not to see correlation coefficients reported for the relationships shown in figures 6–10. Having uncertainty estimates for the coefficients derived from the individual fits included either in the caption or as part of the figure annotations would also be a very sweet addition. (According to page 18, line 7, some of these numbers are given in Table 3.)

Page 17, line 8 : regarding figure 4, the basis for asserting the "smallest particles found at Alta Floresta indicate rather fresh smoke, probably just a few hours after emission" is not immediately apparent. Best I can tell, this is the only example that is not associated with a specific fire, and hence the only example in which the age of the smoke is directly inferred from the size distribution. Since Figure 4 is shown to illustrate "the shift of the size distribution towards larger particles with age of the observed smoke", it seems that have an actual fire (and hence an excellent estimate of age) to associate with the Alta Floresta smoke is reasonably important.

Page 17, line 17 : the caption for Figure 5 should define the 'SPM' acronym. (Should readers assume that SPM is shorthand for 'sun photometer'?)

Page 17, lines 18–19 : Regarding Figure 5, the authors state, "The weak coarse mode may result from aerosols in the boundary layer (marine particles, soil and road dust). The lidar observations do not show this coarse mode." This distinction between full column measurements (AERONET) and range-resolved retrievals (lidar) is well worth emphasizing. When assessing the properties of lofted layers, using AERONET data or parameterizations will always introduce some uncertainties.

Page 17, lines 21–22 : suggest changing "However, in practice, such an approach is not useful" to "However, in all likelihood such an approach would be impractical and/or unreasonably difficult". Separating the contributions from fresh and aged smoke would no doubt be useful. But, as the authors point out, reliably accomplishing the separation would be damnably difficult.

Page 18, line 10 : I wonder what explains the large dispersion of the Table Mountain data seen in Figure 17b?

Page 20, line 27 : perhaps it's worth reminding the reader what the variable $x$ represents? (I had to page back to section 3 to remind myself.)

Page 21, line 6 : first and foremost, the authors deserve a huge round of applause and thanks for the addition of Table 4.

Page 21, line 6 : having 'backscatter lidar' appear twice in the column headers for Table 4 is incredibly confusing. Yes, the explanation is given in the caption. But I strongly believe that it's worth using up a bit of extra page real estate to clear differentiate between ground-based and space-based backscatter lidars in the column headers.

Page 21, line 14 : The authors quite rightly call out the Achilles Heel of elastic backscatter lidar retrievals: "The lidar ratio is even required as input in the basic determination of the backscatter coefficient profiles." This point is, I believe, well worth emphasizing in this manuscript.

Page 21, line 16 : "we assume an uncertainty of 25% in Table 4". Uncertainty estimated for the CALIPSO aerosol backscatter coefficients are given in the aerosol profile products. Their calculation is described in Young et al., 2013 and in the supplementary material for Young et al., 2018.

Page 24, line 14 : regarding Figure 14, and in my role as reviewer #2, I'm disappointed that the authors chose to exclude the third, stratospheric smoke plume from this figure. I doubt other readers will know or care.

Page 24, lines 20–26 : I have several comments on this paragraph.

a) I'm pleased to see the multiple scattering issue specifically acknowledged. For dense aerosol layers, multiple scattering can (and, no doubt, does) introduce substantial unquantified error into the CALIOP retrievals of particulate backscatter and extinction coefficients (e.g., Wandinger et al., 2010; Liu et al., 2011). And to answer a question posed by the authors in their 'responses to the reviewers', I too am of the opinion that the multiple scattering impact in the case of opaque smoke layers is largely unknown. However, the opaque smoke layers identified by the authors in the initial version of their manuscript may offer an opportunity to begin quantifying this impact (albeit very crudely at first).

b) I'm disappointed that only one of the two smoke plumes shown in Figure 14 is further analyzed in Figure 15. In my opinion, the authors are missing an opportunity to establish some practical limits on the application of their method.

c) In the fourth line in the caption for Figure 15, I suggest changing "The CALIPSO backscatter coefficients…" to "The CALIPSO aerosol backscatter coefficients…".

Page 25, line 27 : the DOI for the CALIPSO aerosol profile products is 10.5067/CALIOP/CALIPSO/LID_L2_05KMAPRO-STANDARD-V4-20 (see https://asdc.larc.nasa.gov/project/CALIPSO/CAL_LID_L2_05kmAPro-Standard-V4-20_V4-20)

One final comment: the authors' responses to the reviewers (https://acp.copernicus.org/preprints/acp-2020-1093/acp-2020-1093-AC1-supplement.pdf) claims that "Sections 1-3 are not changed". But even a cursory scan of the revised manuscript shows that this is not exactly true. In fact, the original section 3 ("POLIPHON method: smoke retrieval") has been entirely replaced by a newly titled ("Methodological background: Microphysical properties from backscatter coefficients") and substantially revamped section 3. In my view, this is welcome change.

**References**

Liu, Z., D. Winker, A. Omar, M. Vaughan, C. Trepte, Y. Hu, K. Powell, W. Sun, B. Lin, 2011: Effective lidar ratios of dense dust layers over North Africa derived from the CALIOP measurements, *JQSRT*, **112**, 204–213, https://doi.org/10.1016/j.jqsrt.2010.05.006.

Prata, A. T., S. A. Young, S. T. Siems, and M. J. Manton, 2017: Lidar ratios of stratospheric volcanic ash and sulfate aerosols retrieved from CALIOP measurements, *Atmos. Chem. Phys.*, **17**, 8599–8618, https://doi.org/10.5194/acp-17-8599-2017.

Wandinger, U., M. Tesche, P. Seifert, A. Ansmann, D. Müller, and D. Althausen, 2010: Size matters: Influence of multiple scattering on CALIPSO light-extinction profiling in desert dust, *Geophys. Res. Lett.*, **37**, L10801, https://doi.org/10.1029/2010GL042815.

Young, S. A., M. A. Vaughan, R. E. Kuehn, and D. M. Winker, 2013: The Retrieval of Profiles of Particulate Extinction from Cloud-Aerosol Lidar Infrared Pathfinder Satellite Observations (CALIPSO) Data: Uncertainty and Error Sensitivity Analyses, *J. Atmos. Oceanic Technol.*, **30**, 395–428, https://doi.org/10.1175/JTECH-D-12-00046.1.

Young, S. A., M. A. Vaughan, J. L. Tackett, A, Garnier, J. B. Lambeth, and K. A. Powell, 2018: Extinction and Optical Depth Retrievals for CALIPSO's Version 4 Data Release, *Atmos. Meas. Tech.*, **11**, 5701–5727, https://doi.org/10.5194/amt-11-5701-2018.